# xRFM: Accurate, scalable, and interpretable feature learning models for tabular data

**Daniel Beaglehole**
Computer Science and Engineering
Halıcıoğlu Data Science Institute
UC San Diego
dbeaglehole@ucsd.edu

**David Holzmüller**
INRIA Saclay
david.holzmuller@inria.fr

**Adityanarayanan Radhakrishnan**
MIT Mathematics
Broad Institute of MIT and Harvard
aradha@mit.edu

**Mikhail Belkin**
Halıcıoğlu Data Science Institute
UC San Diego
mbelkin@ucsd.edu

## Abstract

Inference from tabular data, collections of continuous and categorical variables organized into matrices, is a foundation for modern technology and science. Yet, in contrast to the dramatic developments in the rest of AI, the best practice for these predictive tasks has been relatively unchanged and is still primarily based on variations of Gradient Boosted Decision Trees (GBDTs). Very recently, there has been renewed interest in developing state-of-the-art methods for tabular data based on recent developments in neural networks and feature learning methods. In this work, we introduce xRFM, an algorithm that combines feature learning kernel machines with a tree structure to both adapt to the local structure of the data and scale to essentially unlimited amounts of training data.

On the TALENT benchmark, we show that compared to 31 other methods, including recently introduced tabular foundation models (TabPFNv2) and GBDTs, xRFM achieves best performance across 100 regression datasets and is competitive to the best methods across 200 classification datasets outperforming GBDTs. Additionally, xRFM provides interpretability natively through the Average Gradient Outer Product. Code for xRFM (following a scikit-learn-style API) is available at: https://github.com/dmbeaglehole/xRFM.

## 1 Introduction

Tabular data – collections of continuous and categorical variables organized into matrices – underlies all aspects of modern commerce and science from airplane engines to biology labs to bagel shops. Yet, while methods for Machine Learning and AI in language and vision have seen unprecedented progress, the primary methodologies of prediction from tabular data have been relatively static, dominated by variations of Gradient Boosted Decision Trees (GBDTs), such as XGBoost (Chen & Guestrin, 2016). Nevertheless, hundreds of tabular datasets have been assembled to form extensive regression and classification benchmarks (Bischl et al., 2025; Grinsztajn et al., 2022; Fernández-Delgado et al., 2014; Ye et al., 2024; Erickson et al., 2025), and, recently, there has been renewed interest in building state-of-the-art predictive models for tabular data (Hollmann et al., 2025; Holzmüller et al., 2024; Gorishniy et al., 2025). Notably, given the remarkable effectiveness of large, "foundation" models for text, there has been much excitement in developing similar models on tabular data, and recent effort has led to the development of TabPFN-v2, a foundation model for tabular data appearing in Nature (Hollmann et al., 2025). Yet, despite this progress, building scalable, effective, and interpretable machine learning models in this domain remains an open challenge.

In this work, we introduce xRFM, a tabular predictive model that combines recent advances in feature learning kernel machines with an adaptive tree structure, making it highly accurate, scalable, and interpretable. xRFM builds upon the Recursive Feature Machine (RFM) algorithm from Radhakr-

ishnan et al. (2024), which enabled *feature learning* (a form of supervised dimensionality reduction) in general machine learning models. xRFM works as follows: Given training data, it first builds a binary tree structure to split data into subsets based on features relevant for prediction within each split. When splits reach a certain size (of $\leq C$ data samples), we train a *leaf* RFM (a hyper-parameter and compute optimized version of the original RFM).

There are two important consequences of using a tree structure in conjunction with leaf RFMs. The first is that it enables *local feature learning*, meaning that xRFM can learn different features for different subsets of the data. This property is crucial for tackling hierarchical structure in tabular data (for example, one set of features may be relevant when a specific feature takes on high value while another set of features is relevant when the same feature takes low values). The second consequence is that it enables xRFM to scale nearly linearly in the number of samples ($O(n \log n)$ given $n$ samples) during training and logarithmically ($O(\log n)$) during inference. As such, the inference time of xRFM is comparable to that of tree-based models.

In practice, we show xRFM has the best performance across 100 tabular regression tasks and is competitive with state-of-the-art on 200 tabular classification tasks from the TALENT benchmark (Ye et al., 2024). In particular, xRFM outperforms 31 other methods on these regression tasks including the GBDT and neural network variations mentioned in the opening paragraph. On the TabArena-Lite benchmark (Erickson et al., 2025), we show xRFM achieves one of the best tradeoffs (is on the empirical Pareto frontier) between performance and inference time among all methods in regression and is again competitive on classification. We show that xRFM achieves similar results on the largest datasets from the meta-test benchmark (Holzmüller et al., 2024), where directly solving standard kernel machines becomes intractable on standard GPUs. An additional benefit of xRFM is that it natively provides interpretability by exposing features learned and used for prediction. In particular, each leaf RFM learns features through a mathematical object known as the Average Gradient Outer Product (AGOP), whose diagonal indicates coordinates relevant for prediction and whose top eigenvectors indicate directions in data most relevant for prediction. Using examples of xRFM trained on synthetic and real data, we show how AGOP matrices at each leaf RFM shed light on features relevant for prediction, including cases when leaf RFMs learn different features for different splits of data.

To summarize, xRFM combines feature learning kernel methods with adaptive data partitioning using the AGOP. It is a fast, effective, and interpretable model on tabular data at all scales.

## 2 Preliminaries

We begin with a review of kernel machines and kernel-RFM, which we use to build xRFM.

### 2.1 Kernel machines

A kernel machine is a non-parametric machine learning model (Schölkopf & Smola, 2002; Aronszajn, 1950). The idea behind kernel machines is that a nonlinear predictive model can be trained by first transforming input data with a fixed, nonlinear feature map and then performing linear regression on the transformed data. Kernel machines make this procedure computationally tractable even for infinite dimensional feature maps by using explicitly defined kernel functions (inner products of feature mapped data). We describe kernel machines in the context of supervised learning below. Let $X \in \mathbb{R}^{n \times d}$ denote training inputs with $x^{(i)^T}$ denoting the example in the $i^{\text{th}}$ row of $X$ for $i \in [n]$ and $y \in \mathbb{R}^{n \times c}$ denote training labels. Let $K : \mathbb{R}^d \times \mathbb{R}^d \to \mathbb{R}$ denote a kernel function (a positive semi-definite, symmetric function). Given a regularization parameter $\lambda \geq 0$, a kernel machine trained on the data $(X, y)$ is a predictor, $\widehat{f} : \mathbb{R}^d \to \mathbb{R}^c$, of the form:

$$\widehat{f}(x) = K(x, X)\alpha \qquad ; \qquad \alpha = (K(X, X) + \lambda I)^{-1} y ; \qquad (1)$$

where the notation $K(x, X) \in \mathbb{R}^{1 \times n}$ denotes an $n$-dimensional row vector with $K(x, X)_i = K(x, x^{(i)})$ and $K(X, X) \in \mathbb{R}^{n \times n}$ denotes a matrix with $K(X, X)_{ij} = K(x^{(i)}, x^{(j)})$. Examples of kernel functions used in practice are the Gaussian kernel ($K(x, z) = \exp(-\|x - z\|_2^2)/L^2$) or the Laplace kernel ($K(x, z) = \exp(-\|x - z\|_2)/L$). The advantage of kernel functions is that the predictor admits a closed form solution, which can be computationally efficient to fit on datasets under 100k samples. The limitations of kernel machines are twofold: (1) the formulation in Eq. (1) does not scale well (it is super-quadratic in the number of samples) and (2) the choice of kernel is

fixed independently of the input data. The former limitation is addressed by recent, pre-conditioned gradient descent methods for solving for $\alpha$ in Eq. (1) and methods that construct low-dimensional approximations of the kernel matrix. These include EigenPro (Ma & Belkin, 2017; 2019; Abedsoltan et al., 2023), Falkon (Rudi et al., 2017), and randomly pivoted Cholesky (Chen et al., 2025). The latter limitation is addressed by procedures for adapting the kernel to the training data. We describe one such approach known as the kernel Recursive Feature Machine (kernel-RFM) below.

## 2.2 RECURSIVE FEATURE MACHINES (RFMs)

The ability to learn task-relevant features from data is key to building effective predictors (Damian et al., 2022; Ghorbani et al., 2021; Abbe et al., 2022). RFM, introduced in Radhakrishnan et al. (2024), is an algorithm that enables feature learning in general machine learning models through a mathematical object known as the Average Gradient Outer Product (AGOP). Given a predictive model $\widehat{f} : \mathbb{R}^d \to \mathbb{R}$ and data $S = \{x^{(1)}, \ldots, x^{(n)}\} \subset \mathbb{R}^d$, the AGOP is defined as

$$\mathrm{AGOP}(\widehat{f}, S) = \frac{1}{n} \sum_{i=1}^{n} \nabla \widehat{f}(x^{(i)}) \nabla \widehat{f}(x^{(i)})^T \in \mathbb{R}^{d \times d}, \tag{2}$$

where $\nabla \widehat{f}(x^{(i)})$ denotes the gradient of $\widehat{f}$ at the point $x^{(i)}$. The AGOP is an estimate of the (uncentered) covariance of the gradients of $\widehat{f}$ and intuitively captures the subspace along which the predictor highly varies (Trivedi et al., 2014; Kpotufe et al., 2016). The RFM algorithm involves iterating between training a predictive model and using the AGOP of the trained model to select features and linearly transform input data. As such, feature learning through RFM can be viewed as a form of "supervised" PCA.

In the particular case of kernel machines, which have no native mechanism for feature learning, RFM enables feature learning by adapting the kernel to the data. We describe the kernel-RFM algorithm below. Following the notation in the previous section, let $(X, y)$ denote training inputs and labels, let $K$ denote the kernel function, and $\lambda$ denote the regularization parameter. Letting $M_1 = I$ and $c > 0$, kernel-RFM repeats the following two steps for $T$ iterations:

Step 1: $\widehat{f}_t(x) = K(M_t x, X M_t)\alpha_t \qquad ; \qquad \alpha_t = [K(X M_t, X M_t) + \lambda I]^{-1} y,$

Step 2: $M_{t+1} = \left[ \mathrm{AGOP}(\widehat{f}_t(M_t x), X) \right]^c. \tag{3}$

Typical choices of $T$ and $c$ in practice are $T \leq 10$ and $c \in \{\frac{1}{4}, \frac{1}{2}\}$ (Radhakrishnan et al., 2024; Beaglehole et al., 2025; 2024; Mallinar et al., 2024). (In practice, we return the $f_t$ with best validation performance rather than returning $f_T$.) When training labels, $y$, are multi-dimensional, we use the Jacobian of $\widehat{f}$ instead of the gradient in Step (2) (i.e., averaging the AGOP over output dimensions). Kernel-RFM is particularly effective at identifying low dimensional subspaces (or subsets of variables) relevant for prediction (Radhakrishnan et al., 2025; Zhu et al., 2025), making it a useful interpretability tool, as we will show in Section 5.

## 3 XRFM ALGORITHM OVERVIEW

We now describe our algorithm xRFM, which consists of the following two components (Fig. 1):

(1) An improved kernel-RFM, termed *leaf* RFM, that is trained on the subset of data;

(2) A binary tree that splits the data into subsets, termed *leaves*, of a maximum size (*leaf size*) that is independent of the number of training samples. Leaf RFMs are then trained on the subset at each leaf. The splits stratify data based on features relevant for prediction.

Together these components enable xRFM to perform local feature learning (learning different features on different subpopulations) and achieve $O(n \log n)$ training time as well as $O(\log n)$ inference time given $n$ samples. We outline these two components in detail below.

## 3.1 LEAF RFM

Here, we describe the changes we made to the original kernel RFM algorithm from Radhakrishnan et al. (2024) to produce leaf RFMs. The kernel RFM model was primarily built using kernels

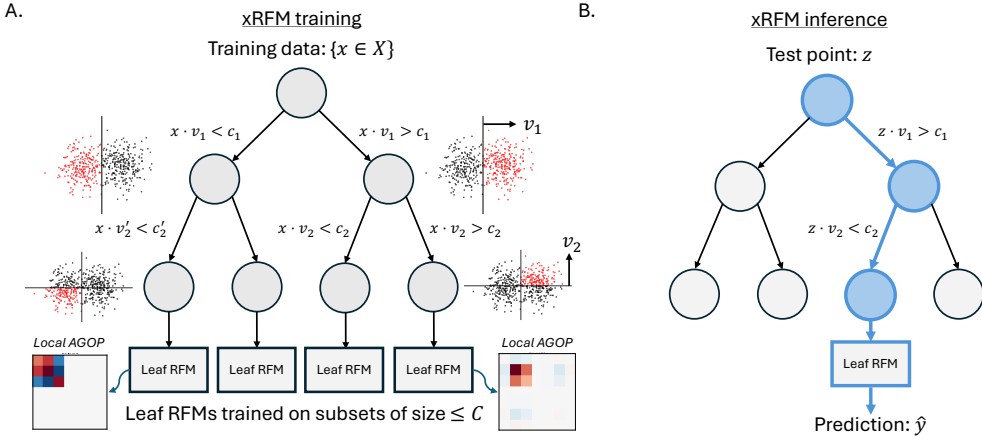

**Figure 1:** Overview of xRFM training and inference procedures. (A) xRFM is trained by splitting the data along the median projections (denoted $c_1, c_2$) onto computed split directions (denoted $v_1, v_2$). Data is split repeatedly into leaves, which contain at most $C$ training samples. Leaf RFMs are trained on the data at each leaf. (B) During inference, test data is routed to the appropriate leaf RFM based on split directions. The prediction is generated by the selected leaf RFM.

that were invariant to orthonormal transformations of data (those where $K(x, z) = K(Ux, Uz)$ for any orthogonal matrix $U$). Examples include the Gaussian kernel of the form $K(x, z) = \exp(-L\|x - z\|_2^2)$ or the Laplace kernel of the form $K(x, z) = \exp(-L\|x - z\|_2)$. Such invariance does not effectively leverage a special property of tabular data – the fact that each coordinate can be independently meaningful. This property has been hypothesized as an explanation for the superior performance of tree-based models, which compose functions of individual coordinates, over traditional neural networks on tabular datasets (Grinsztajn et al., 2022). To account for this special structure, we introduce the following modifications to kernel-RFM:

(1) We tune over a more general class of kernels $K_{p,q}(x, x') = \exp(-\|x - x'\|_p^q / L^q)$ that are positive definite for $0 < q \le p \le 2$ (Schoenberg, 1942, Theorems 1, 5).

(2) We tune over using the full AGOP and just the diagonal of the AGOP. The latter is known to be a theoretically grounded approach for coordinate selection (Zhu et al., 2025), and introduces an axis-aligned bias that has been observed to match the structure of tabular data (Grinsztajn et al., 2022).

In addition to the above changes, we also implemented optimizations to speed up computations for categorical variables and an adaptive approach for tuning bandwidth separately for data on each leaf. Additional details regarding these changes and hyperparameter search spaces for leaf RFMs on TALENT, TabArena-Lite, and meta-test are provided in Appendix A.

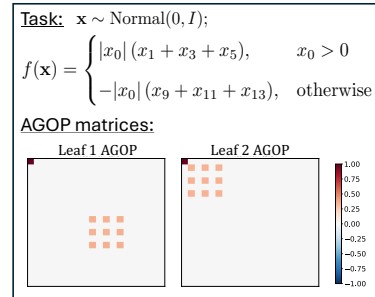

### 3.2 TREE-BASED DATA PARTITIONING

We now explain how xRFM builds a binary tree structure to partition data into leaves on which leaf RFMs are trained. First, given a dataset $S$ with $n$ samples, we subsample $m$ points and train a leaf RFM (referred to as a *split* model) for one iteration on this subsample. The split model serves to learn a direction that can be used to partition the data into two subsets. To this end, we extract the top eigenvector, $v$, of the AGOP from the split model and create two subsets of the $n$ datapoints: $S_1 = \{x \in S \; ; \; v^T x > \text{Median}(v^T z \text{ for } z \in S)\}$ and $S_2 = \{x \in S \; ; \; v^T x \le \text{Median}(v^T z \text{ for } z \in S)\}$. We repeat this procedure on $S_1$ and $S_2$ and

**Figure 2:** Training xRFM on synthetic data where splitting on the top AGOP direction enables xRFM to learn locally relevant features.

their corresponding children until all leaves are less than a maximum leaf size $C$. We finally train a leaf RFM on each of these leaves. This procedure is illustrated in Fig. 1A and detailed in the 'TreePartition' procedure of Algorithm A.2.

Note that our splitting procedure results in leaves of equal sizes (because of the split based on median), unlike tree-based predictive models, which can result in uneven splits. We typically split large datasets until leaves contain at most 60,000 samples. Leaf RFM hyperparameters are tuned using only on the validation data that falls into the leaf (as determined by splits down the binary tree). In the case that validation data at a given leaf has too few samples, we further hold out a subset of the training data at the leaf to use for validation.

Note that it is possible to have split data based on unsupervised splitting procedures (e.g., those based on random projections or using top eigenvectors of PCA) (Dasgupta & Freund, 2008). A key advantage of our split approach is that it groups together data points based on the features most relevant for prediction, as is captured by the top eigenvector of the AGOP. We found that splitting in this manner outperformed splits based on unsupervised approaches, including those described above. We include a discussion and comparison of split methods in Appendix B and Tables E.1, E.2. Prior works have also split data using the random forest procedure. For example, Hollmann et al. (2025) train TabPFN-v2 on the samples routed to a given leaf of a tree in a random forest. The procedure in xRFM differs as it stratifies samples by projection onto a direction rather than individual coordinates and produces only one balanced tree instead of a forest.

**Advantages of tree splits over traditional methods for scaling kernels.** Because of tree-based partitioning, xRFM is able to scale log-linearly in the number of samples during training[1] and logarithmically in the number of samples during inference. While this is a computationally appealing aspect in practice, we note that there are a number of existing methods for scaling kernel machines. Examples include the Nyström approximation (Williams & Seeger, 2000) and fast preconditioner methods for solving kernel regression (Ma & Belkin, 2017; Rudi et al., 2017; Ma & Belkin, 2019; Abedsoltan et al., 2023).

A major advantage of xRFM over these other methods and kernel RFM itself is the ability for tree-based partitioning to learn features local to subsets of data. Here, in Fig. 2, we provide an example clearly demonstrating this benefit. In this example, the function $f : \mathbb{R}^d \to \mathbb{R}$ depends on coordinates $x_1, x_3, x_5$ if $x_0 > 0$ and depends on coordinates $x_9, x_{11}, x_{13}$ if $x_0 \leq 0$. xRFM learns $x_0$ as the initial split direction (through the AGOP of the split model trained on a random subset), then learns the relevant features for each case in separate leaf RFM models. In contrast, if running kernel RFM from Radhakrishnan et al. (2024), the model would simply return that $\{x_0, x_1, x_3, x_5, x_9, x_{11}, x_{13}\}$ are all relevant features without decomposing into the two cases.

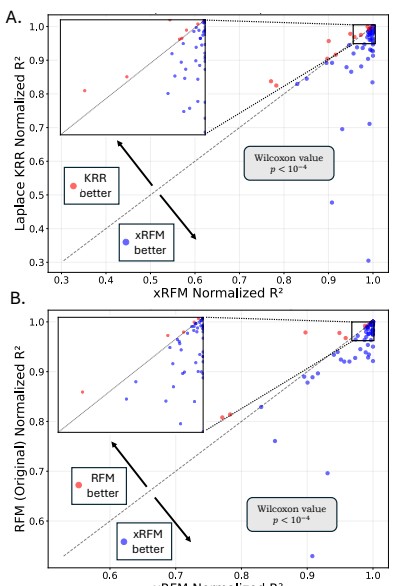

**Figure 4:** Comparisons of xRFM with (A) kernel ridge regression and (B) the original RFM (Radhakrishnan et al., 2025) on TALENT regression datasets (metric is normalized $R^2$, see Appendix A). Each point is a dataset.

## 4    XRFM PERFORMANCE

We now apply xRFM to three tabular data benchmarks: the TALENT benchmark (Ye et al., 2024), the TabArena-Lite benchmark (Erickson et al., 2025), and large datasets from the meta-test benchmark (Holzmüller et al., 2024). We use TALENT and TabArena-Lite to evaluate xRFM on datasets of various sizes between 500 and 150,000 training samples.[2] We use the meta-test benchmark to

---

[1]The complexity for the applications of RFM itself scales only linearly, but splitting the data with the median at every tree node increases the complexity to $O(n \log n)$.

[2]We use the "Lite" version of TabArena, as the full version is computationally expensive due to at least $9\times$ more outer folds.

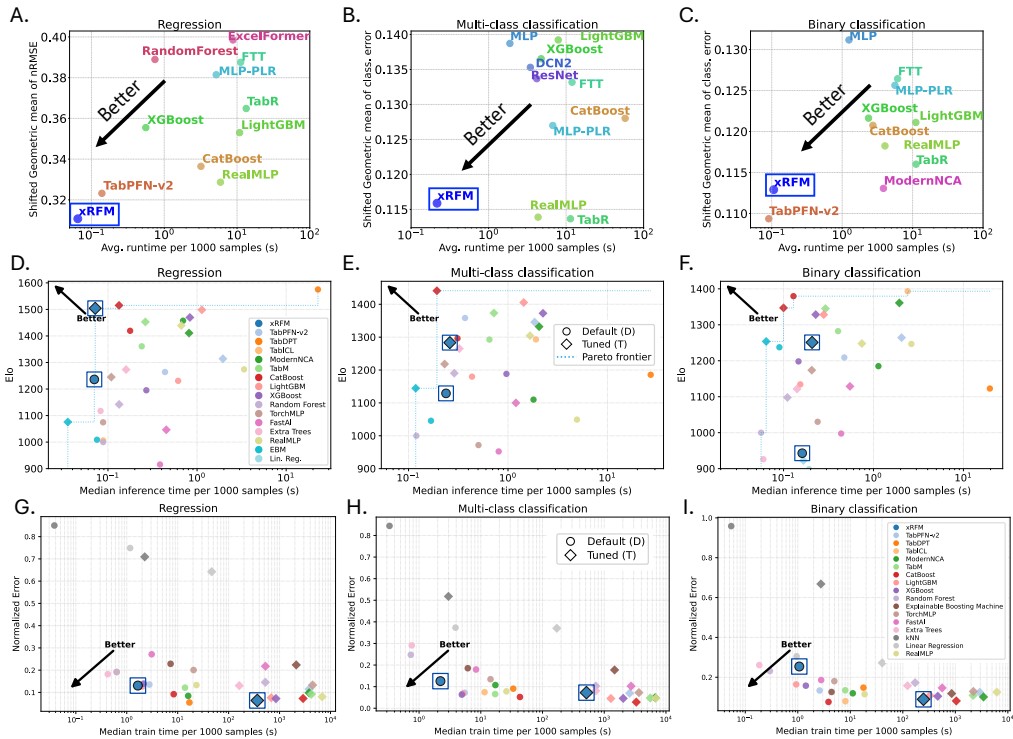

**Figure 3:** Performance and runtime of xRFM on the TALENT (Plots A-C) and TabArena-Lite benchmarks (Plots D-I). The y-axes of plots A-C are the shifted geometric mean of the error across all datasets in that category, while the x-axes are the average over all datasets of the training plus inference time per 1000 samples for just the best hyperparameter configuration (meaning if a dataset has $n$ samples, we compute the training and inference time on the $n$ samples divide the total time by $n/1000$). The y-axes in plots D-F are Elo, the main metric used in TabArena, and reflect the relative win-rate of each method, while the x-axes are the median inference time per 1000 total samples. Plots G-I show normalized error versus median train time per 1000 total samples on TabArena-Lite. The TabArena-Lite plots D-I additionally show the default methods and plots D-F show the Pareto tradeoff curve for inference time versus Elo. The TALENT plots are (A) nRMSE over 100 regression datasets, (B) classification error over 80 multi-class datasets, (C) classification error over 120 binary classification datasets. The TabArena-Lite plots are (D,G) regression, (E,H) multi-class, and (F,I) binary datasets. For multiclass and binary classification plots H and I, the TabArena benchmark computes the error metric as log-loss and $1$-AUROC, respectively.

evaluate xRFM on larger datasets with at least $70,000$ and up to $500,000$ samples (a setting in which direct linear solvers become intractable, taking more than $40$GB VRAM).

**Performance evaluation measures.** When measuring performance on these datasets, we use Root Mean Square Error (RMSE) for regression tasks (as is used in TALENT), and we use classification error ($1 -$ classification accuracy) for classification tasks. For TabArena-Lite, binary classification performance for individual datasets are measured by AUROC, while multi-class datasets are evaluated with log-loss. To measure performance on aggregate over TALENT, we consider the following aggregation metrics on individual dataset performance measures: Shifted Geometric Mean, Arithmetic Mean, and Normalized Arithmetic Mean. TabArena-Lite uses Elo (Elo, 1967), a rating system that calculates the relative skill levels of each method based on the probability of winning when pairs of methods are compared on individual datasets (see Erickson et al. (2025) for how Elo is estimated in this benchmark). All other metrics are defined in Appendix A.

**Results on the TALENT and TabArena-Lite benchmarks.** The TALENT benchmark consists of 300 total datasets comparing 31 different supervised learning algorithms (Ye et al., 2024). Among these 31 algorithms are strong, widely used predictive models including Gradient Boosted Decision Tree (GBDT) variants (like XGBoost, CatBoost, LightGBM), hyper-parameter optimized neural networks (RealMLP), and recent transformer-based foundation models (TabPFN-v2). The TALENT benchmark consists of 100 regression tasks, 120 binary classification tasks, and 80 multiclass classification tasks. The number of training samples per task varies between 500 and 100, 000 and each task contains at least 5 input variables. TabArena-Lite contains 51 total datasets (13 regression and 38 classification) comparing 15 different supervised learning algorithms including many of those in TALENT. The datasets in TabArena-Lite vary in sample size from 700 to 150, 000.

xRFM is the best performing method on TALENT regression tasks according to all aggregation metrics over RMSEs on individual datasets (Fig. 3A). It is also the fastest method per configuration, although TabPFN-v2 does not incur a $100\times$ overhead for hyperparameter tuning. xRFM is also competitive on classification datasets (Fig. 3B, C). Namely, xRFM is the third highest ranked method on classification tasks. xRFM also achieves win-rates significantly higher than 50% when compared to state-of-the-art neural networks and GBDT models on the TALENT benchmark (Fig. F.1). We additionally compare the performance (in Elo) and inference time of xRFM using the TabArena-Lite benchmark (Fig. 3). In particular, when compared to all default and tuned methods for this benchmark, tuned xRFM is among the top three methods for regression while being orders of magnitude faster for inference. Indeed, xRFM lies along the empirical Pareto frontier, meaning that there is no method that dominates xRFM in both performance and inference time for these tasks (Fig. 3D). For classification tasks, xRFM is near the empirical Pareto frontier (Fig. 3E, F). We additionally show that when comparing the mean normalized error metric (rather than ELO, which is based on winrate), tuned xRFM outperforms all other tuned and default methods except for the TabDPT foundation model on regression tasks, is among the top three methods on binary classification tasks, and is competitive on multi-class classification tasks (Fig. 3G-I). We also compare xRFM performance against that of kernel ridge regression and standard RFM from Radhakrishnan et al. (2024). xRFM is significantly better than these models (p-value $< 10^{-4}$, Wilcoxon test, Fig. 4).

Fig. 5 shows that the overall training+inference cost of xRFM initially scales super-linearly. However, xRFM is still reasonably fast at 60K samples, after which the log-linear scaling thanks to the tree-based data partitioning kicks in (Fig. F.2). Notably, xRFM is extremely fast on small datasets: it is at least two orders of magnitude faster than other methods when the dataset contains fewer than 3000 samples (we omit TabPFN-v2 in our comparison as it could not be run on all datasets due to restrictions on the allowable sample size, number of features, and output dimensionality).

**Results on large datasets from the meta-test benchmark.** To analyze xRFM performance on large datasets beyond those in TALENT and TabArena, we consider the 17 largest datasets in the meta-test benchmark from Holzmüller et al. (2024) (Table E.9).

We compare the performance of xRFM to other models with results reported in the literature on these 17 datasets (Fig. 6). These other models include GBDTs (XGBoost, LightGBM, CatBoost), and neural networks (ResNet, MLP, MLP-PLR (Gorishniy et al., 2022), RealMLP). We report the percentage improvement of each method over MLPs, following the procedure in Gorishniy et al. (2025); Qu et al. (2025). Similar to the results on TALENT, xRFM is best on regression tasks, and second best on classification tasks. All results are presented in Tables E.9, E.10. For meta-test, we search over slightly larger regularization parameters and slightly smaller bandwidth parameters to adapt to the much larger sample sizes of these datasets.

## 5 FEATURES LEARNED BY XRFM ON TABULAR DATA

An advantage of xRFM is that it immediately provides a means of identifying features relevant for prediction (without stacking on any additional interpretability methods). Namely, we can extract learned AGOPs of leaf RFMs and visualize the features they select.

We use two approaches for identifying features selected by the AGOP. The first is to identify the elements along the diagonal of the AGOP with highest magnitude. By definition, these entries indicate how much a leaf RFM's predictions vary when perturbing a given coordinate. As such, they are a natural measure of feature importance. The second approach is to examine the loadings onto

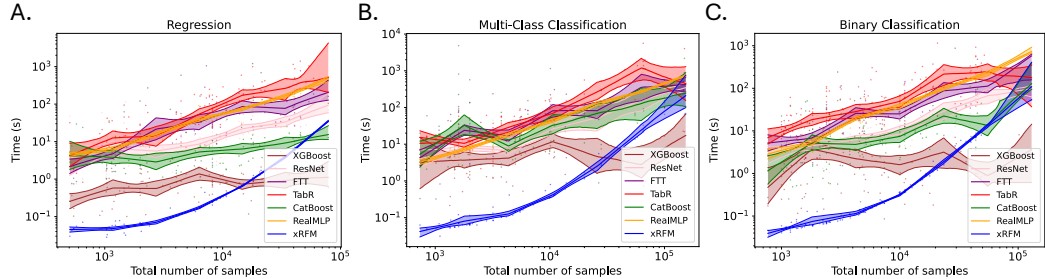

**Figure 5:** Total training and inference time for the best hyperparameter configuration as a function of the number of samples (training+validation+testing) across the TALENT benchmark. Curves indicate piece-wise linear fit to measures on each dataset (shown as points). (A) Results across 100 regression tasks. (B) Results across 80 multi-class classification tasks. (C) Results across 120 binary classification tasks.

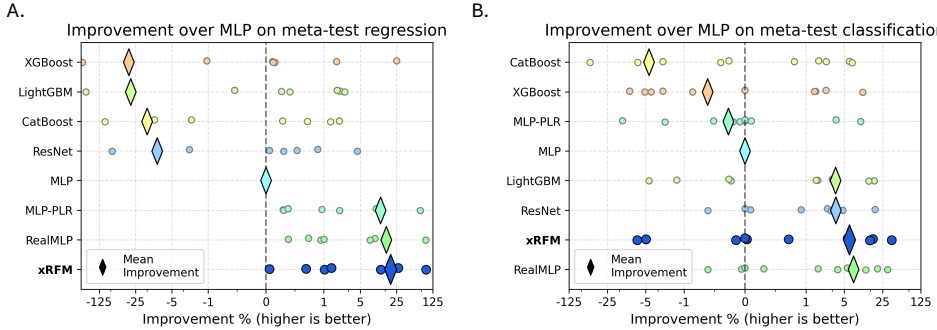

**Figure 6:** Performance comparison across the 17 large datasets from meta-test (70,000-500,000 samples). (A, B) Percentage improvement over MLP error on (A) regression and (B) classification datasets. Average percentage improvement is denoted as a diamond. Points denote individual dataset results (points are jittered for visibility).

the top eigenvectors of the AGOP. This approach allows us to identify joint effects of feature perturbations on the prediction. Namely, if coordinate $i$ of the top eigenvector is positive and coordinate $j$ is negative, then increasing one of these features and decreasing the other changes the prediction.

We use both approaches above to understand what features are learned by xRFM on tabular data from scikit-learn and meta-test (Fig. 7). As examples, we study the AGOPs for four such datasets: (1) California housing - a regression task for predicting the average price of a house, (2) Covertype - a multiclass classification task for identifying the dominant tree species in a given location, (3) NYC Taxi Tipping - a regression task for predicting the dollar tip amount in a taxi ride, and (4) Breast cancer - a classification task for identifying malignancy from features of biopsy images.

As the California housing and breast cancer datasets contain fewer than 50k samples (the parameter we used for leaf size), we visualize a single AGOP. For covertype, we visualize one of the AGOPs from a leaf RFM (for this dataset, the leaf RFM AGOPs generally indicate the same pattern), and for taxi tipping we visualize several leaf RFM AGOPs. Upon visualization, it is apparent that AGOPs indicate low rank structure: either they highlight a subset of coordinates relevant for prediction (Fig. 7A, B) or they have a decay in the eigenvalue spectrum (Fig. 7C).

In Fig. 7A and B, we list feature names for the features with highest diagonal AGOP entries (darker shades of red indicate higher values). In all cases, we find that AGOP identifies sensible features for prediction. For the California housing dataset, xRFM identifies longitude, or east-west location, as the most important feature for predicting the average price of a house. Given that beach fronts (and major cities) in California are typically located to the west, the importance of house longitude is consistent with the hypothesis that homes closer to the beach are more expensive on average. For

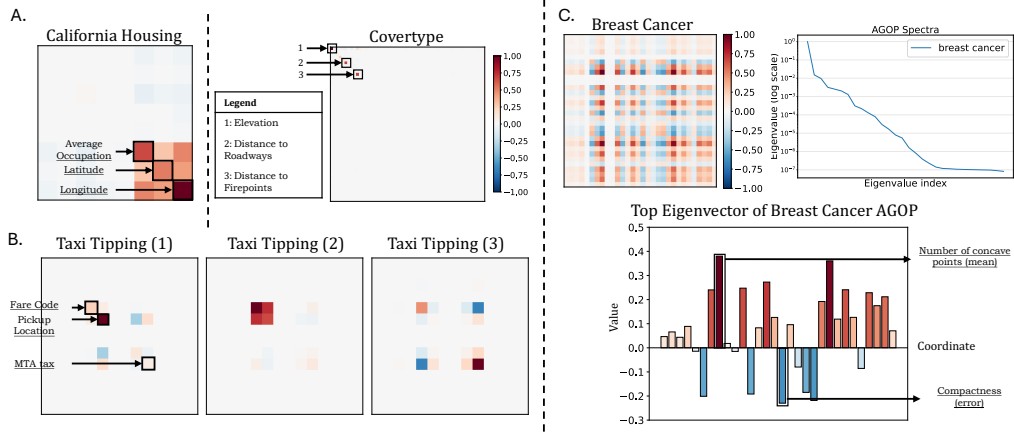

**Figure 7:** Interpreting xRFM through the AGOP of its constituent Leaf RFM models. (A) Examining the most important features for xRFM trained on California Housing (price prediction) and Covertype (dominant tree species prediction) datasets, based on the magnitude of diagonal entries. (B) Examining the features identified across three different Leaf RFM models for the NYC Taxi Tipping dataset. (C) Examining features learned for Breast Cancer detection from processed FNA imaging. The spectrum of this AGOP is plotted and the top eigenvector is shown in a bar plot. The most positive and negative entries of this eigenvector are boxed.

the Covertype prediction dataset, the example AGOP from a leaf RFM shows that elevation, distance to roadways, and distance to firepoints are the most important features. This finding is consistent with the hypothesis that elevations with different climates and the existence of fires / roadways can significantly affect viability of different tree species. For the taxi tipping dataset, we observe that leaf RFMs identify varying local features. For example, one leaf RFM (denoted Taxi Tipping 1 in Fig. 7B) selected pickup location as an important feature, while this feature was less important for a different leaf RFM (Taxi Tipping 3). Furthermore, fare code and the MTA tax have varying feature relationships at each leaf: in leaf RFM 1 and 2, the fare code and MTA tax has neutral or synergistic effects on the prediction value, while for leaf RFM 3, increasing fare code has the opposite effect on prediction as increasing MTA tax (shown as a blue square in Fig. 7B).

For the breast cancer dataset, the most important feature found by xRFM is the mean number of concave points in the biopsy image, which has been shown to be a significant indicator of malignancy (Narasimha et al., 2013). When examining the top eigenvector of the AGOP, we find that standard error in the compactness measurements of cell nuclei is the feature with highest importance and has an opposite effect on malignancy as concavity (Fig. 7C). Our method suggests that benign cells have less uniform compactness than malignant cells.

## 6 Discussion

Tabular data has historically provided a proving ground for the development of novel predictive models. However, even as the last years have seen spectacular improvements in language processing and computer vision, the progress in tabular data has been far more modest. While this may be changing with the developments such as hyperparameter-optimized neural networks (e.g., RealMLP (Holzmüller et al., 2024)), and very recently, pre-trained transformer-based tabular networks, such as TabPFN-v2 (Qu et al., 2025), there is still large scope for improved tabular data prediction.

Kernel machines provide a powerful conceptually and computationally elegant approach for predictive modeling – one simply transforms data with a nonlinear feature map and then performs linear regression. Yet, historically, these methods have not out-performed Gradient Boosted Decision Trees (GBDTs), while also being significantly more difficult to scale. There are two key limitations of standard kernel machines: (1) standard kernel choices lack adaptivity to features relevant for prediction, and (2) naive implementations scale super-quadratically in the number of training samples,

making them difficult to train on larger datasets (past 70k training samples). Significant progress in overcoming limitation (1) was made in recent work (Radhakrishnan et al., 2024), which introduced Recursive Feature Machines (RFM) to enable feature learning in a range of models, including kernel machines. In fact, feature learning through RFM can be provably more sample efficient than standard kernels (Zhu et al., 2025). There has also been significant work addressing the limitation (2) including (Williams & Seeger, 2000; Ma & Belkin, 2019; Rudi et al., 2017), to reference a few.

In this work, we have taken a different, tree-based approach to overcome these limitations. Our approach enables scalability to large datasets (log-linear training time and logarithmic inference time) while also leveraging the strength of RFM in identifying and exploiting local features. Indeed, the ability of xRFM to learn different features for different subpopulations of data through AGOPs at leaves is useful for characterizing heterogeneity across large datasets. Building on this core novelty and other algorithmic optimizations, we discuss a number of potential improvements and follow ups below.

A direction for future improvement of xRFM is to explore the use of iterative kernel solvers to train leaf RFMs. By using early stopping, these methods could obviate the need for tuning ridge regularization. The use of the adaptive binary tree structure to split data also opens up new avenues for exploration. For instance, it would be interesting to characterize the tradeoff between leaf size and performance and to develop new methods for determining when to stop splitting based statistics of leaf data. It might also be useful to consider alternative bandwidth selection methods for each leaf, for example by optimizing for bandwidths directly as in (He et al., 2023). Lastly, it would be important to understand the effectiveness of AGOP as an interpretability mechanism, particularly in comparison to other widely used methods such as Shapley values (Lundberg & Lee, 2017), gradient-based importance scores (Selvaraju et al., 2017; Shrikumar et al., 2017), and feature importance scores for tree-based models (Breiman et al., 1984).

Overall, we have shown that xRFM is an effective algorithm for inference from tabular data that scales to essentially unlimited data sizes and achieves performance exceeding or comparable to the current state-of-the-art. It combines the advantages of tree-based methods with the power and elegance of feature-learning kernel machines. We envision that xRFM will be used for both high-performing predictive modeling and uncovering heterogeneous structure in large-scale tabular data.

## ACKNOWLEDGMENTS

We would like to thank Jingang Qu, Ingo Steinwart, Tizian Wenzel, and Gaël Varoquaux for relevant discussions. We thank Charles Durham for contributing the KerMac library, which led to drastic speed up in kernel evaluations and gradient computations. We gratefully acknowledge support from the National Science Foundation (NSF) under grants CCF-2112665 and MFAI 2502258, the Office of Naval Research (ONR N000142412631) and the Defense Advanced Research Projects Agency (DARPA) under Contract No. HR001125CE020. This work used the Delta system at the National Center for Supercomputing Applications through allocation TG-CIS220009 from the Advanced Cyberinfrastructure Coordination Ecosystem: Services & Support (ACCESS) program, which is supported by National Science Foundation grants #2138259, #2138286, #2138307, #2137603, and #2138296. We thank Ingo Steinwart for providing computational resources funded by Deutsche Forschungsgemeinschaft (DFG, German Research Foundation) under Germany's Excellence Strategy - EXC 2075 – 390740016.

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

## A METHODS

### A.1 ADDITIONAL LEAF-RFM MODIFICATIONS

Below, we provide further detail on the RFM modifications we made to implement leaf RFM.

(1) We tune over a more general class of kernels $K_{p,q}(x, x') = \exp(-\|x - x'\|_p^q / L^q)$ that are positive definite for $0 < q \leq p \leq 2$ (Schoenberg, 1942, Theorems 1, 5). We typically search over $p \in \mathcal{U}(0.7, 1.4)$ and $p \in \{q, 2\}$ (Table A.1).

(2) We tune over using the full AGOP and just the diagonal of the AGOP. The latter is known to be a theoretically grounded approach for coordinate selection Zhu et al. (2025), and introduces an axis-aligned bias that has been observed to match the structure of tabular data (Grinsztajn et al., 2022).

We also make the following changes to improve leaf RFM runtime on tabular data and to allow xRFM to adapt to the variance of input variables at each leaf:

(3) We speed up computation for kernels with $q = 1$ on categorical variables taking $c$ values by precomputing possible kernel entries as follows. We restrict the AGOP matrix to be block diagonal with a block for the $c \times c$ entries corresponding to the categorical variable. Letting $M_c \in \mathbb{R}^{c \times c}$ denote this block and $e_i \in \{0, 1\}^c$ denote the one-hot embedding of the categorical variable when it takes on value $i$, we precompute $M_c^{1/2}(e_i - e_j)$ for all $i, j \in [c]$.

(4) We tune over whether or not to use *adaptive bandwidth*, which involves separately scaling $L$ at each leaf RFM by $(\text{Median}(\|x - x'\|_p)))^{-1}$ for $x \neq x'$ at every leaf independently. Adaptive scaling allows xRFM to adapt to the variance of covariates at different leaves.

### A.2 METRICS

Following Holzmüller et al. (2024), we report the shifted geometric mean of the error, which is the geometric mean of the error after shifting by a small value to prevent over-sensitivity to datasets with small errors. This metric is defined as follows.

**Definition 1.** *For a given set of errors on a benchmark $\varepsilon_1, \ldots, \varepsilon_N$, the shifted geometric mean ($\text{SGM}_\varepsilon$) with parameter $\varepsilon$ takes value:*

$$\text{SGM}_\varepsilon = \exp\left(\frac{1}{N} \sum_{i=1}^{N} \log(\varepsilon + \varepsilon_i)\right) .$$

*In this work, as in Holzmüller et al. (2024), we use $\varepsilon = 0.01$.*

For regression tasks, we use normalized Root-mean-square-error (nRMSE), which is defined as follows.

**Definition 2.** *The normalized Root-mean-square-error is defined as,*

$$nRMSE = \sigma_y^{-1} \sqrt{\frac{1}{N} \sum_{i=1}^{N} (y_i - \widehat{y}_i)^2}, \qquad where \ \sigma_y = \sqrt{\frac{1}{N} \sum_{i=1}^{N} (y_i - \bar{y})^2}$$

When we refer to other normalized metrics, such as those used in Table E.4, Table E.5, Table E.6, Table E.7, Table E.8, we are min-max normalizing errors across methods for each dataset (see below).

**Definition 3.** *The normalized error $\widetilde{E}_j$ for method $j$ on a given dataset, where the un-normalized error is $E_j$, has the form:*

$$\widetilde{E}_j = \frac{E_j - E_{\min}}{E_{\max} - E_{\min}}, \qquad E_{\min} = \min_k E_k, \quad E_{\max} = \max_k E_k .$$

### A.3 HYPERPARAMETERS

| Hyperparameter | TALENT | TabArena-Lite | Meta-test |
|---|---|---|---|
| Bandwidth | $\log \mathcal{U}(1, 200)$ | $\log \mathcal{U}(0.5, 200)$ | $\log \mathcal{U}(0.4, 80)$ |
| Bandwidth Mode | {constant} | {constant} | {constant, adaptive} |
| Categorical Transformations | {one_hot} | {one_hot} | {ordinal_encoding, one_hot} |
| Diagonal | {False, True} | {False, True} | {False, True} |
| Early Stop Multiplier | 1.1 | 1.1 | 1.1 |
| Exponent $q$ | $\mathcal{U}(0.7, 1.4)$ | $\mathcal{U}(0.7, 1.4)$ | $\mathcal{U}(0.7, 1.3)$ |
| Kernel Type | {80%: $K_{p,q}$, 20% $K_{2,q}$} | {80%: $K_{p,q}$, 20% $K_{2,q}$} | {80%: $K_{p,q}$, 20% $K_{2,q}$} |
| $p$ (when kernel type is $K_{p,q}$) | $\mathcal{U}(q, q + 0.8(2 - q))$ | $\mathcal{U}(q, q + 0.8(2 - q))$ | $\mathcal{U}(q, q + 0.8(2 - q))$ |
| Normalization | {standard} | {standard} | {standard} |
| Regularization | $\log \mathcal{U}(10^{-6}, 1)$ | $\log \mathcal{U}(10^{-6}, 10)$ | $\log \mathcal{U}(10^{-5}, 50)$ |
| Refill size ($N_{\text{val}}$) | 1500 | 1500 | 1500 |
| Min. split subset size ($C$) | 60,000 | 60,000 | 60,000 |

**Table A.1:** Search spaces for xRFM on the TALENT (Figure 3), TabArena-Lite (Figure 3), and Meta-test benchmarks (Figure 6).

**Data pre-processing.** For the TALENT benchmark, each method's hyperparameters are tuned after fixing a single data normalization and categorical encoding scheme. For xRFM, kernel ridge regression, and standard kernel RFM, we one-hot encode categorical features and z-score input coordinates separately. For meta-test, methods also tune over choice of ordinal or one-hot encoding of categorical variables. On meta-test, we also normalize data by z-scoring input coordinates separately prior to tuning xRFM parameters.

**Details of various methods.** For scaling kernel ridge regression and the standard kernel RFM to large TALENT datasets, we used Eigenpro-2 (EP2) (Ma & Belkin, 2019) that was initialized with the coefficients obtained from directly solving RFM on a random sample of 70,000 points. To tune the optimization hyperparameters for EP2, we tuned the model parameters for 100 trials on a random 70,000 sample subset, then for the final run, we used the tuned hyperparameters (and for kernel RFM the AGOP from the best iteration). For TabPFN-v2, we used the code provided from the benchmark github (https://github.com/LAMDA-Tabular/TALENT), which subsampled to 10,000 samples to avoid out-of-memory issues. The provided code did not apply TabPFN-v2 to datasets with more than 10 classes. For efficient $L_p^q$ kernel computations on GPU we used the KerMac library (https://github.com/Kernel-Machines/kermac).

**Split method and TabPFN-v2 embeddings experimental details**   For the experiments evaluating choices of split methods, we tune among hard-routing (0 temperatures) and 15 additional temperatures at even spacings in log space from 0.025 to 4.5. We use a minimum split subset size of 40,000 samples. For the experiments on learning from embeddings of TabPFN-v2, we tune hyperparameters for xRFM (on embeddings or the tabular data itself) by sampling 50 random configurations of the TabArena search space. For preprocessing, we treat the embeddings from TabPFN-v2 as numeric inputs and apply standard scaling to each coordinate.

## A.4   ALGORITHMIC DETAILS

**Note on AGOP computation**   For computing gradients of the predictor on training data (for AGOP computation), we omit the contribution to the gradient from the kernel evaluation between each training point and itself. This is because the kernel function is often not differentiable (e.g. Laplace kernels) when evaluated for two identical points.

---

**Algorithm A.1** Leaf RFM

**Require:**
- $x^{(1)}, \ldots, x^{(n)} \in \mathbb{R}^d, y \in \mathbb{R}^{n \times c}$ : Train data
- $X_{\text{val}} \in \mathbb{R}^{m \times d}, y_{\text{val}} \in \mathbb{R}^{m \times c}$ : Validation data
- $K(\cdot, \cdot; L, p)$: Kernel parametrized by bandwidth $L \in \mathbb{R}^+$ and exponent $p \in (0, 2]$
- $\tau \in \mathbb{Z}^+$: Number of iterations
- $\lambda \in \mathbb{R}^+$: Ridge parameter
- $\varepsilon \in \mathbb{R}^+$: Stability parameter
- use_diag: Boolean indicating whether to use diagonal of AGOP only
- adapt_bandwidth: Boolean indicating whether to adapt bandwidth

$M_0 \leftarrow I_{d \times d}$
$X = [x^{(1)}, \ldots, x^{(n)}]^\top \in \mathbb{R}^{n \times d}$
**if** adapt_bandwidth **then**
    $L \leftarrow \text{AdaptBandwidth}(L)$                            ▷ Adapt bandwidth if enabled
**end if**
**for** $t = 0, \ldots, \tau - 1$ **do**
    **if** use_diag **then**
        $X_M \leftarrow X \odot \text{diag}(M_t)^{1/2}$
        Solve $\alpha_t$ such that $(K(X_M, X_M) + \lambda I)\alpha_t = y$
        $f^{(t)}(x) = K(x \odot \text{diag}(M_t)^{1/2}, X_M)\alpha_t$      ▷ $\odot$ denotes element-wise multiplication
    **else**
        $X_M \leftarrow X M_t^{1/2}$
        Solve $\alpha_t$ such that $(K(X_M, X_M) + \lambda I)\alpha_t = y$
        $f^{(t)}(x) = K(M_t^{1/2} x, X_M)\alpha_t$
    **end if**
    Compute $E_t \leftarrow \text{Error}(f^{(t)}, X_{\text{val}}, y_{\text{val}})$                  ▷ Validate model
    $M_{t+1} \leftarrow \frac{1}{n} \sum_{i=1}^n \nabla_x f^{(t)}(x^{(i)}) \nabla_x f^{(t)}(x^{(i)})^\top \in \mathbb{R}^{d \times d}$      ▷ Feature matrix (AGOP) computation
    $M_{t+1} \leftarrow M_{t+1} / (\varepsilon + \max_{i,j} M_{t+1}[i, j])$            ▷ Normalize feature matrix
**end for**
$t^* \leftarrow \arg\min_t E_t$
**return** $\alpha_{t^*}, M_{t^*}$      ▷ KRR coefficients: $\alpha_{t^*}$, feature matrix: $M_{t^*}$ from best iteration on val. set

---

---

**Algorithm A.2** TreePartition

---

**Require:**
- $\mathcal{D} = \{(x^{(1)}, y^{(1)}), \ldots, (x^{(n)}, y^{(n)})\}$ : Full dataset with $x^{(i)} \in \mathbb{R}^d, y^{(i)} \in \mathbb{R}^c$
- $K(\cdot, \cdot; L, p)$: Kernel parametrized by bandwidth $L \in \mathbb{R}^+$ and exponent $p \in (0, 2]$
- $N \in \mathbb{Z}^+$: Number of sample points for Leaf RFM
- $L \in \mathbb{Z}^+$: Maximum leaf size
- $\lambda \in \mathbb{R}^+$: Ridge parameter

**function** TREEPARTITION($\mathcal{D}$)
    **if** $|\mathcal{D}| \leq L$ **then**
        **return** Leaf node with dataset $\mathcal{D}$
    **end if**
    Sample $N$ points $\mathcal{S} = \{(x^{(a_1)}, y^{(a_1)}), \ldots, (x^{(a_N)}, y^{(a_N)})\}$ from $\mathcal{D}$
    $X_s = [x^{(a_1)}, \ldots, x^{(a_N)}]^\top \in \mathbb{R}^{N \times d}$
    $y_s = [y^{(a_1)}, \ldots, y^{(a_N)}]^\top \in \mathbb{R}^{N \times c}$
    Solve $\alpha$ such that $(K(X_s, X_s) + \lambda I)\alpha = y$             ▷ Fit Leaf RFM on sampled data
    Define predictor $f(x) = K(x, X_s)\alpha$
    Compute AGOP: $M \leftarrow \frac{1}{N} \sum_{i=1}^{N} \nabla_x f(x^{(a_i)}) \nabla_x f(x^{(a_i)})^\top \in \mathbb{R}^{d \times d}$
    Extract top eigenvector $v_1$ of $M$             ▷ Principal direction
    Project all data points: $p^{(i)} \leftarrow v_1^\top x^{(i)}$ for $i = 1, \ldots, |\mathcal{D}|$
    Compute median projection: $m \leftarrow \text{Median}(\{p^{(1)}, \ldots, p^{(|\mathcal{D}|)}\})$
    Split dataset:
        $\mathcal{D}_{\text{left}} \leftarrow \{(x^{(i)}, y^{(i)}) \in \mathcal{D} : v_1^\top x^{(i)} \leq m\}$
        $\mathcal{D}_{\text{right}} \leftarrow \{(x^{(i)}, y^{(i)}) \in \mathcal{D} : v_1^\top x^{(i)} > m\}$
    left_child $\leftarrow$ TreePartition($\mathcal{D}_{\text{left}}$)             ▷ Recursive call
    right_child $\leftarrow$ TreePartition($\mathcal{D}_{\text{right}}$)           ▷ Recursive call
    **return** Internal node with splitting vector $v_1$, threshold $m$, and children
**end function**

---

**Algorithm A.3** Route (*find the leaf that contains a point*)

---

**Require:**
- $\mathcal{T}$ : a (possibly trained) tree whose internal nodes store
    - splitting vector $v_1 \in \mathbb{R}^d$
    - threshold $m \in \mathbb{R}$
- $x \in \mathbb{R}^d$ : query point

**function** ROUTE($x, \mathcal{T}$)
    $r \leftarrow \mathcal{T}.\text{root}$             ▷ Initialize current node $r$ from the tree
    **while** $r$ is an internal node **do**
        **if** $v_1^\top x \leq r.\text{threshold}$ **then**     ▷ Check if projection is less than (median) threshold
            $r \leftarrow r.\text{left\_child}$
        **else**
            $r \leftarrow r.\text{right\_child}$
        **end if**
    **end while**
    **return** $r$             ▷ $r$ is now the leaf node that contains $x$
**end function**

---

---

**Algorithm A.4** xRFM (*training*)

---

**Require:**
- $\mathcal{D}_{\text{train}}$, $\mathcal{D}_{\text{val}}$ : training and validation sets
- $K(\cdot,\cdot; L, p)$ : kernel (bandwidth $L$, exponent $p$)
- TreeHyp $= \{N, L_{\max}, \lambda\text{split}\}$ : hyper-parameters for TREEPARTITION
- LeafHyp $= \{\tau, \lambda_{\text{leaf}}, \varepsilon, use\_diag\}$ : hyper-parameters for LEAFRFM

 

**function** XRFM-FIT($\mathcal{D}_{\text{train}}, \mathcal{D}_{\text{val}}$)
    $\mathcal{T} \leftarrow$ TREEPARTITION($\mathcal{D}_{\text{train}}, K$, TreeHyp)            $\triangleright$ Alg. A.2
    **for all** leaf node $\ell \in$ LEAVES($\mathcal{T}$) **do**
        $\mathcal{D}_\ell^{\text{train}} \leftarrow$ data stored in $\ell$
        $\mathcal{D}_\ell^{\text{val}} \leftarrow \{(x,y)\in\mathcal{D}_{\text{val}} :$ ROUTE$(x, \mathcal{T}) = \ell\}$
        $(\alpha_\ell, M_\ell) \leftarrow$ LEAFRFM($\mathcal{D}_\ell^{\text{train}}, \mathcal{D}_\ell^{\text{val}}, K$, LeafHyp)       $\triangleright$ Alg. A.1
        Define predictor $f_\ell(x)$ according to $(\alpha_\ell, M_\ell)$ and store it in $\ell$
    **end for**
    **return** $\mathcal{T}$            $\triangleright$ Tree whose leaves now carry trained predictors
**end function**

---

**Algorithm A.5** xRFM (*inference*)

---

**Require:**
- $\mathcal{T}$ : trained tree returned by XRFM-FIT (Alg. A.4)
- $x \in \mathbb{R}^d$ : test point

**function** XRFM-PREDICT($\mathcal{T}, x$)
    $\ell \leftarrow$ ROUTE($x, \mathcal{T}$)            $\triangleright$ Alg. A.3
    $\widehat{y} \leftarrow f_\ell(x)$            $\triangleright$ Leaf predictor stored in $\ell$
    **return** $\widehat{y}$
**end function**

---

## B    Optimizing the xRFM tree structure

Here we discuss two avenues for optimizing the tree structure of xRFM: (1) ensembling Leaf RFM models at prediction time, and (2) selecting the method for extracting split directions.

We may expect for some datasets that the target function being estimated exhibits a relatively simple parametric form, e.g. a fixed quadratic function of the input features. In these cases, or in cases where the choices of split are poor, splitting the data across leaves may worsen performance. To account for this, we also consider routing test samples at prediction time to multiple leaf RFM models, then forming the prediction by taking a weighted average of the predictions across these models. We implement this weighted averaging and demonstrate its improved performance, at the expense of larger evaluation times and slightly larger training times.

**Temperature tuning (TT).**    Our ensembling procedure works by assigning the weight to each leaf equal to the product of node-wise weights at each node along the path to that leaf. The weight for node $i$, with a given split threshold $c_i$ and split direction $v_i$, is equal $\sigma\left(\frac{v_i \cdot x - c_i}{\beta_i}\right)$, where $\beta_i$ is a scaling factor chosen for each node. We set the scaling factor $\beta_i = B \cdot \mathrm{IQR}_i$, where $B$ is a temperature parameter and $\mathrm{IQR}_i$ is the inter-quartile range of value for $v_i \cdot z$, where $z$ are all the training samples whose paths to their leaves include that node. As this temperature parameter can be tuned independently of training the leaf RFM models, we simply fit all the leaf RFM models individually, then optimize the temperature parameter by grid-search on the validation set. Note that the "hard-routed" xRFM model, which deterministically routes each test point to a single leaf, is equivalent to this "soft-routed" xRFM model with $B = 0$. For additional inference-time speed-up, we allow only the top 8 highest weighted leaves to participate in the ensemble, and re-normalize the weights so that they sum to 1 across participating leaves for each test point.

**Split methods.**    In addition to using the top eigenvector of the AGOP on a split model, we consider two additional splitting choices: (1) Unsupervised splits (PCA) using the top principal component of the training samples at each node, and (2) Random forest importances (RF), where the coordinate with maximal information gain is selected at every node.

We evaluate temperature tuning and split methods on the large meta-test datasets where at least 2-3 splits are required for every dataset using a minimum split size of 40,000 samples. Our results show that supervised splits outperform unsupervised ones, (Tables E.1,E.2). Further, while AGOP splitting gives the fastest method, using AGOP or RF splitting along with temperature tuning gave the best test error.

## C    Meta-feature analysis

We examine the effect of dataset selection and metafeatures on the relative performance of xRFM. First, we examine whether xRFM is better on the TALENT or TabArena datasets. In Fig.E.14, we show that xRFM is the best method on regression using the TALENT baselines and tuning procedure, even when restricted to the datasets in TALENT that are also in TabArena. Further, xRFM actually improves to the best overall method on binary classification on the TabArena subset, versus the second overall method to TabPFN-v2 on the remaining, non-TabArena subset. Thus, the difference in results between TabArena and TALENT cannot be explained by dataset selection alone. In particular, we show xRFM is the best overall method on regression and binary classification on the subset of datasets in TALENT that is also in the TabArena benchmark (new Tables E.14 and E.15). Other factors that could contribute to the explanation are the use of random search (instead of Optuna), cross-validation, and stronger (more expensive) baselines in TabArena. Moreover, for classification, TabArena uses different metrics (AUC and log-loss), where the log-loss could potentially be improved using post-hoc calibration.

In Fig.F.3, we report the effect of three metafeatures on the normalized errors of xRFM and other top methods. We do not see an obvious trend in the relative performance of xRFM across sample size, feature dimension, or the fraction of features that are categorical.

## D   COMBINING XRFM WITH OTHER MODELS

We also analyzed to what extent xRFM could be combined with other models to give effective ensembles. In Table E.17, we show that performance can be further improved on the TALENT benchmark by including xRFM in a portfolio with CatBoost and XGBoost. In particular, we select the best of xRFM, CatBoost, and XGBoost on the validation set and evaluate this best model on the test set. Note the validation results for TALENT were only provided for 200 of the 300 datasets for CatBoost.

# E  ALL RESULTS

| Dataset | AGOP | Splitting method AGOP + TT | PCA + TT | RF + TT |
|---|---|---|---|---|
| Airlines_DepDelay_10M | 0.9792 | **0.9782** | 0.9785 | 0.9785 |
| Allstate_Claims_Severity | **0.6480** | 0.6509 | 0.6526 | 0.6521 |
| black_friday | 0.6879 | 0.6879 | 0.6891 | **0.6877** |
| Buzzinsocialmedia_Twitter | 0.2300 | 0.2290 | **0.2137** | 0.2138 |
| nyc-taxi-green-dec-2016 | 0.5268 | **0.5250** | 0.6283 | 0.5256 |
| wave_energy | **0.0024** | **0.0024** | **0.0024** | **0.0024** |
| Yolanda | 0.7852 | **0.7790** | 0.7824 | 0.7824 |
| Number of wins: | 2 | **4** | 2 | 2 |
| Shifted geometric mean: | 0.3446 | 0.3440 | 0.3499 | **0.3411** |
| Arithmetic mean: | 0.5514 | 0.5503 | 0.5638 | **0.5489** |
| Average fit time [s]: | **7755** | 8287 | 7765 | 7831 |
| Average eval time [s]: | **77** | 187 | 183 | 174 |

**Table E.1:** Evaluation of split methods on large regression datasets from meta-test. We consider three methods for choosing split directions - AGOP, Principal Component Analysis (PCA), and Random Forest criterion (RF). We also evaluate ensembling leaf RFM models using the temperature tuning method described in Appendix B.

| Dataset | AGOP | Splitting method AGOP + TT | PCA + TT | RF + TT |
|---|---|---|---|---|
| airlines | 0.3318 | 0.3309 | 0.3307 | **0.3289** |
| covertype | 0.0254 | **0.0250** | 0.0262 | 0.0252 |
| Diabetes130US | 0.3856 | **0.3854** | 0.3855 | **0.3854** |
| dionis | **0.0905** | **0.0905** | **0.0905** | **0.0905** |
| Fashion-MNIST | **0.0880** | **0.0880** | **0.0880** | **0.0880** |
| Higgs | 0.2597 | 0.2551 | 0.2548 | **0.2546** |
| jannis | 0.2719 | 0.2718 | 0.2725 | **0.2714** |
| KDDCup99 | 0.0003 | 0.0003 | **0.0002** | 0.0002 |
| kick | **0.0972** | **0.0972** | **0.0972** | **0.0972** |
| MiniBooNE | 0.0539 | **0.0539** | 0.0542 | 0.0545 |
| numerai28.6 | **0.4738** | **0.4738** | **0.4738** | **0.4738** |
| porto-seguro | **0.0382** | **0.0382** | 0.0382 | **0.0382** |
| Number of wins: | 5 | 8 | 5 | **9** |
| Shifted geometric mean: | 0.1159 | **0.1156** | 0.1159 | 0.1156 |
| Arithmetic mean: | 0.1764 | 0.1758 | 0.1760 | **0.1757** |
| Average fit time [s]: | **6410** | 6770 | 6535 | 6510 |
| Average eval time [s]: | **95** | 136 | 127 | 126 |

**Table E.2:** Evaluation of split methods on large classification datasets from meta-test. We consider three methods for choosing split directions - AGOP, Principal Component Analysis (PCA), and Random Forest criterion (RF). We also evaluate ensembling leaf RFM models using the temperature tuning method described in Appendix B.

| Dataset | Task | RFM | xRFM |
|---|---|---|---|
| Credit-c | Multiclass | 0.1395 | **0.1072** |
| Rain-in-Australia | Multiclass | **0.0535** | 0.0629 |
| SDSS17 | Multiclass | 0.0108 | **0.0005** |
| accelerometer | Multiclass | 0.0273 | **0.0190** |
| customer-satisfaction-in-airline | Binary Classification | 0.0152 | **0.0146** |
| dabetes-130-us-hospitals | Binary Classification | **0.0430** | 0.0448 |
| walking-activity | Multiclass | 0.0627 | **0.0167** |
| Arithmetic Mean | – | 0.0503 | **0.0379** |

**Table E.3:** Evaluation of xRFM vs. original RFM on the largest datasets from TALENT, where at least one xRFM split is required. We show the normalized errors (across all 32 methods). Datasets are ordered by training set size.

| Method | Rank | Score | Norm. Score | Top-1 (%) | Top-3 (%) | Top-5 (%) | Top-8 (%) | $SGM_\varepsilon$ |
|---|---|---|---|---|---|---|---|---|
| xRFM | **4.70** | **0.311** | **0.036** | **20.0** | **56.0** | **69.0** | **84.0** | **0.311** |
| TabPFN-v2 | 6.55 | 0.323 | 0.067 | **20.0** | 45.0 | 57.0 | 72.0 | 0.323 |
| CatBoost | 7.79 | 0.336 | 0.053 | 7.00 | 24.0 | 41.0 | 68.0 | 0.336 |
| RealMLP | 8.20 | 0.329 | 0.076 | 8.00 | 21.0 | 37.0 | 60.0 | 0.329 |
| ModernNCA | 9.39 | 0.365 | 0.097 | 14.0 | 30.0 | 38.0 | 57.0 | 0.365 |
| LightGBM | 9.91 | 0.353 | 0.062 | 6.00 | 16.0 | 34.0 | 52.0 | 0.353 |
| XGBoost | 10.6 | 0.355 | 0.071 | 2.00 | 12.0 | 32.0 | 47.0 | 0.355 |
| TabR | 10.7 | 0.365 | 0.133 | 5.00 | 20.0 | 34.0 | 46.0 | 0.365 |
| FTT | 12.0 | 0.388 | 0.149 | 3.00 | 11.0 | 17.0 | 33.0 | 0.388 |
| Laplace KRR | 12.3 | 0.373 | 0.102 | 3.00 | 11.0 | 21.0 | 35.0 | 0.373 |
| MLP-PLR | 12.7 | 0.381 | 0.142 | 0.0 | 8.00 | 15.0 | 34.0 | 0.381 |
| Excelformer | 12.7 | 0.399 | 0.157 | 0.0 | 2.00 | 9.00 | 28.0 | 0.399 |
| PTARL | 13.1 | 0.390 | 0.114 | 2.00 | 6.00 | 9.00 | 20.0 | 0.390 |
| RandomForest | 13.3 | 0.389 | 0.088 | 5.00 | 12.0 | 22.0 | 35.0 | 0.389 |
| AutoInt | 14.3 | 0.399 | 0.158 | 1.00 | 2.00 | 6.00 | 12.0 | 0.399 |
| Node | 14.4 | 0.451 | 0.195 | 0.0 | 6.00 | 18.0 | 31.0 | 0.451 |
| MLP | 15.2 | 0.418 | 0.167 | 1.00 | 3.00 | 4.00 | 11.0 | 0.418 |
| DCN2 | 15.3 | 0.473 | 0.227 | 0.0 | 2.00 | 10.0 | 20.0 | 0.473 |
| ResNet | 15.6 | 0.426 | 0.175 | 0.0 | 4.00 | 8.00 | 15.0 | 0.426 |
| Tangos | 16.3 | 0.432 | 0.173 | 0.0 | 1.00 | 4.00 | 9.00 | 0.432 |
| SNN | 17.2 | 0.419 | 0.174 | 0.0 | 2.00 | 4.00 | 8.00 | 0.419 |
| kNN | 19.1 | 0.459 | 0.190 | 2.00 | 4.00 | 4.00 | 11.0 | 0.459 |
| TabNet | 21.5 | 0.475 | 0.227 | 0.0 | 0.0 | 0.0 | 0.0 | 0.475 |
| GrowNet | 23.0 | 0.619 | 0.323 | 0.0 | 0.0 | 0.0 | 0.0 | 0.619 |
| SVM | 23.2 | 0.612 | 0.346 | 0.0 | 0.0 | 1.00 | 1.00 | 0.612 |
| TabTransformer | 26.4 | 1.07 | 0.750 | 1.00 | 1.00 | 2.00 | 3.00 | 1.07 |
| SwitchTab | 26.9 | 1.29 | 0.745 | 0.0 | 0.0 | 0.0 | 1.00 | 1.29 |
| Danets | 27.2 | 1.12 | 0.830 | 0.0 | 1.00 | 1.00 | 3.00 | 1.12 |

**Table E.4:** Full TALENT Regression results across 100 datasets. Rank is the average rank among the ordered methods over all datasets. Score is the metric we use to compare methods in Figure 3, in this case $SGM_\epsilon$. Normalized score is the arithmetic mean of the normalized nRMSE. Top-X (%) is the percentage of datasets for which that method is in the top X ranks. The final column is the shifted geometric mean error ($SGM_\varepsilon$).

| Method | Rank | Score | Norm. Score | Top-1 (%) | Top-3 (%) | Top-5 (%) | Top-8 (%) | $SGM_\varepsilon$ |
|---|---|---|---|---|---|---|---|---|
| TabPFN-v2 | **7.15** | 0.823 | **0.935** | **25.0** | **51.5** | **63.2** | **70.6** | 0.108 |
| xRFM | 7.60 | 0.825 | 0.933 | 11.8 | 29.4 | 48.5 | 66.2 | 0.107 |
| RealMLP | 7.64 | 0.823 | 0.918 | 10.3 | 20.6 | 48.5 | 66.2 | 0.107 |
| TabR | 7.98 | **0.828** | 0.932 | 8.82 | 27.9 | 41.2 | 60.3 | **0.106** |
| ModernNCA | 8.88 | 0.825 | 0.912 | 10.3 | 30.9 | 44.1 | 61.8 | 0.106 |
| CatBoost | 10.3 | 0.819 | 0.905 | 1.47 | 19.1 | 36.8 | 51.5 | 0.117 |
| LightGBM | 11.7 | 0.813 | 0.883 | 5.88 | 22.1 | 33.8 | 44.1 | 0.125 |
| XGBoost | 12.1 | 0.815 | 0.886 | 4.41 | 19.1 | 26.5 | 41.2 | 0.124 |
| ResNet | 12.3 | 0.807 | 0.879 | 0.0 | 4.41 | 16.2 | 36.8 | 0.127 |
| FTT | 12.8 | 0.810 | 0.868 | 0.0 | 5.88 | 10.3 | 29.4 | 0.125 |
| MLP-PLR | 13.2 | 0.815 | 0.881 | 0.0 | 7.35 | 11.8 | 25.0 | 0.119 |
| Laplace KRR | 13.5 | 0.804 | 0.864 | 5.88 | 16.2 | 20.6 | 33.8 | 0.133 |
| DCN2 | 14.0 | 0.806 | 0.865 | 1.47 | 4.41 | 10.3 | 26.5 | 0.127 |
| MLP | 14.2 | 0.805 | 0.864 | 0.0 | 4.41 | 8.82 | 19.1 | 0.131 |
| SNN | 14.9 | 0.805 | 0.858 | 0.0 | 0.0 | 4.41 | 8.82 | 0.129 |
| AutoInt | 15.7 | 0.804 | 0.858 | 0.0 | 0.0 | 1.47 | 7.35 | 0.130 |
| RandomForest | 15.7 | 0.797 | 0.831 | 5.88 | 8.82 | 13.2 | 27.9 | 0.137 |
| Excelformer | 16.1 | 0.805 | 0.853 | 0.0 | 0.0 | 4.41 | 17.6 | 0.127 |
| Tangos | 16.5 | 0.797 | 0.844 | 0.0 | 1.47 | 4.41 | 13.2 | 0.133 |
| TabCaps | 16.5 | 0.797 | 0.837 | 0.0 | 1.47 | 5.88 | 13.2 | 0.134 |
| Danets | 17.7 | 0.795 | 0.829 | 0.0 | 1.47 | 2.94 | 5.88 | 0.137 |
| PTARL | 19.1 | 0.796 | 0.808 | 0.0 | 1.47 | 1.47 | 4.41 | 0.139 |
| kNN | 20.1 | 0.786 | 0.772 | 2.94 | 7.35 | 10.3 | 14.7 | 0.152 |
| LogReg | 20.2 | 0.767 | 0.755 | 1.47 | 4.41 | 7.35 | 13.2 | 0.162 |
| TabTransformer | 21.5 | 0.771 | 0.739 | 0.0 | 1.47 | 1.47 | 2.94 | 0.155 |
| Node | 22.0 | 0.767 | 0.742 | 0.0 | 4.41 | 7.35 | 10.3 | 0.173 |
| SVM | 23.4 | 0.750 | 0.697 | 2.94 | 5.88 | 5.88 | 8.82 | 0.182 |
| GrowNet | 24.4 | 0.691 | 0.596 | 0.0 | 0.0 | 2.94 | 5.88 | 0.216 |
| SwitchTab | 25.0 | 0.748 | 0.674 | 0.0 | 1.47 | 1.47 | 2.94 | 0.187 |
| TabNet | 26.0 | 0.753 | 0.615 | 0.0 | 0.0 | 0.0 | 0.0 | 0.183 |
| NaiveBayes | 29.4 | 0.609 | 0.327 | 0.0 | 0.0 | 0.0 | 0.0 | 0.321 |
| NCM | 30.5 | 0.627 | 0.278 | 0.0 | 0.0 | 1.47 | 1.47 | 0.326 |

**Table E.5:** Full TALENT Multiclass classification ($\leq 10$ classes) results. Rank is the average rank among the ordered methods over all datasets. (Normalized) Score is the metric we use to compare methods in Figure 3. In this case score is the mean classification accuracy (1 minus the error in Figure 3). Top-X (%) is the percentage of datasets for which that method is in the top X ranks. The final column is the shifted geometric mean error ($SGM_\varepsilon$).

| Method | Rank | Score | Norm. Score | Top-1 (%) | Top-3 (%) | Top-5 (%) | Top-8 (%) | $SGM_\varepsilon$ |
|---|---|---|---|---|---|---|---|---|
| RealMLP | **3.75** | **0.730** | **0.964** | 8.33 | **50.0** | **91.7** | **91.7** | **0.164** |
| xRFM | 5.46 | 0.718 | 0.949 | 25.0 | 50.0 | 66.7 | 83.3 | 0.183 |
| TabR | 6.00 | 0.720 | 0.948 | 16.7 | 50.0 | 50.0 | 66.7 | 0.172 |
| ResNet | 6.33 | 0.720 | 0.949 | 0.0 | 16.7 | 50.0 | 83.3 | 0.178 |
| ModernNCA | 6.83 | 0.713 | 0.913 | **33.3** | 41.7 | 50.0 | 83.3 | 0.168 |
| FTT | 8.79 | 0.709 | 0.919 | 0.0 | 8.33 | 25.0 | 50.0 | 0.187 |
| MLP-PLR | 8.79 | 0.712 | 0.925 | 0.0 | 0.0 | 25.0 | 50.0 | 0.185 |
| MLP | 9.58 | 0.710 | 0.922 | 0.0 | 0.0 | 8.33 | 41.7 | 0.190 |
| Laplace KRR | 10.8 | 0.679 | 0.889 | 0.0 | 16.7 | 41.7 | 50.0 | 0.218 |
| DCN2 | 11.2 | 0.707 | 0.915 | 0.0 | 8.33 | 8.33 | 8.33 | 0.193 |
| AutoInt | 12.2 | 0.701 | 0.899 | 0.0 | 0.0 | 0.0 | 16.7 | 0.194 |
| SNN | 12.2 | 0.704 | 0.906 | 0.0 | 0.0 | 0.0 | 33.3 | 0.197 |
| CatBoost | 12.8 | 0.679 | 0.842 | 8.33 | 25.0 | 33.3 | 33.3 | 0.218 |
| Danets | 15.9 | 0.687 | 0.882 | 0.0 | 0.0 | 0.0 | 0.0 | 0.215 |
| Tangos | 16.1 | 0.660 | 0.823 | 0.0 | 8.33 | 8.33 | 16.7 | 0.225 |
| XGBoost | 16.2 | 0.678 | 0.874 | 8.33 | 8.33 | 8.33 | 25.0 | 0.236 |
| Excelformer | 16.6 | 0.675 | 0.851 | 0.0 | 0.0 | 0.0 | 16.7 | 0.217 |
| TabCaps | 16.8 | 0.670 | 0.853 | 0.0 | 0.0 | 0.0 | 0.0 | 0.231 |
| LightGBM | 16.9 | 0.668 | 0.850 | 0.0 | 0.0 | 0.0 | 16.7 | 0.261 |
| PTARL | 18.7 | 0.656 | 0.823 | 0.0 | 0.0 | 0.0 | 0.0 | 0.233 |
| kNN | 20.2 | 0.637 | 0.800 | 0.0 | 0.0 | 0.0 | 8.33 | 0.273 |
| RandomForest | 20.7 | 0.620 | 0.782 | 0.0 | 8.33 | 8.33 | 8.33 | 0.311 |
| TabTransformer | 20.8 | 0.582 | 0.702 | 0.0 | 0.0 | 8.33 | 8.33 | 0.307 |
| LogReg | 22.0 | 0.538 | 0.622 | 0.0 | 0.0 | 0.0 | 0.0 | 0.338 |
| SVM | 25.1 | 0.486 | 0.523 | 0.0 | 8.33 | 8.33 | 8.33 | 0.385 |
| SwitchTab | 25.2 | 0.525 | 0.603 | 0.0 | 0.0 | 0.0 | 0.0 | 0.372 |
| Node | 26.2 | 0.536 | 0.611 | 0.0 | 0.0 | 0.0 | 0.0 | 0.402 |
| GrowNet | 26.2 | 0.402 | 0.477 | 0.0 | 0.0 | 0.0 | 0.0 | 0.565 |
| TabNet | 26.3 | 0.486 | 0.516 | 0.0 | 0.0 | 0.0 | 0.0 | 0.368 |
| NaiveBayes | 29.5 | 0.410 | 0.380 | 0.0 | 0.0 | 0.0 | 0.0 | 0.545 |
| NCM | 30.2 | 0.399 | 0.339 | 0.0 | 0.0 | 0.0 | 0.0 | 0.552 |

**Table E.6:** Full TALENT Multiclass classification ($> 10$ classes) results. Rank is the average rank among the ordered methods over all datasets. (Normalized) Score is the metric we use to compare methods in Figure 3. In this case score is the mean classification accuracy (1 minus the error in Figure 3). Top-X (%) is the percentage of datasets for which that method is in the top X ranks. The final column is the shifted geometric mean error ($SGM_\varepsilon$).

| Method | Rank | Score | Norm. Score | Top-1 (%) | Top-3 (%) | Top-5 (%) | Top-8 (%) | $SGM_\varepsilon$ |
|---|---|---|---|---|---|---|---|---|
| xRFM | **5.96** | **0.845** | **0.981** | **29.6** | **55.6** | **59.3** | 74.1 | 0.119 |
| RealMLP | 7.44 | 0.839 | 0.951 | 7.41 | 14.8 | 29.6 | **77.8** | 0.125 |
| TabR | 7.83 | 0.844 | 0.970 | 18.5 | 33.3 | 55.6 | 70.4 | **0.116** |
| ModernNCA | 9.69 | 0.843 | 0.963 | 3.70 | 29.6 | 40.7 | 59.3 | 0.120 |
| CatBoost | 10.2 | 0.827 | 0.891 | 7.41 | 25.9 | 33.3 | 55.6 | 0.128 |
| LightGBM | 10.5 | 0.826 | 0.884 | 7.41 | 18.5 | 40.7 | 55.6 | 0.128 |
| TabPFN-v2 | 10.9 | 0.834 | 0.923 | 7.41 | 22.2 | 25.9 | 48.1 | 0.129 |
| MLP-PLR | 11.2 | 0.827 | 0.890 | 0.0 | 0.0 | 18.5 | 40.7 | 0.131 |
| XGBoost | 12.1 | 0.824 | 0.875 | 3.70 | 18.5 | 33.3 | 51.9 | 0.129 |
| FTT | 12.4 | 0.829 | 0.900 | 0.0 | 11.1 | 18.5 | 29.6 | 0.135 |
| MLP | 12.7 | 0.833 | 0.915 | 0.0 | 7.41 | 14.8 | 29.6 | 0.130 |
| ResNet | 13.3 | 0.834 | 0.920 | 0.0 | 3.70 | 7.41 | 29.6 | 0.130 |
| DCN2 | 13.4 | 0.832 | 0.919 | 0.0 | 3.70 | 3.70 | 22.2 | 0.130 |
| PTARL | 13.4 | 0.831 | 0.911 | 0.0 | 3.70 | 7.41 | 14.8 | 0.131 |
| AutoInt | 13.7 | 0.831 | 0.911 | 0.0 | 0.0 | 3.70 | 11.1 | 0.130 |
| Excelformer | 14.7 | 0.821 | 0.858 | 3.70 | 7.41 | 7.41 | 11.1 | 0.143 |
| SNN | 14.8 | 0.831 | 0.912 | 0.0 | 3.70 | 7.41 | 18.5 | 0.131 |
| Laplace KRR | 17.0 | 0.826 | 0.879 | 3.70 | 11.1 | 14.8 | 14.8 | 0.138 |
| Danets | 17.4 | 0.827 | 0.887 | 0.0 | 0.0 | 0.0 | 0.0 | 0.133 |
| TabCaps | 18.4 | 0.822 | 0.862 | 0.0 | 3.70 | 7.41 | 7.41 | 0.136 |
| TabTransformer | 19.2 | 0.816 | 0.796 | 0.0 | 3.70 | 7.41 | 18.5 | 0.160 |
| RandomForest | 19.6 | 0.806 | 0.772 | 0.0 | 3.70 | 7.41 | 18.5 | 0.144 |
| Node | 20.1 | 0.793 | 0.724 | 3.70 | 3.70 | 3.70 | 7.41 | 0.151 |
| LogReg | 20.5 | 0.807 | 0.778 | 0.0 | 3.70 | 18.5 | 25.9 | 0.159 |
| Tangos | 20.7 | 0.811 | 0.798 | 0.0 | 0.0 | 3.70 | 3.70 | 0.139 |
| SVM | 21.8 | 0.802 | 0.754 | 0.0 | 0.0 | 3.70 | 18.5 | 0.162 |
| GrowNet | 22.7 | 0.804 | 0.771 | 0.0 | 0.0 | 0.0 | 3.70 | 0.160 |
| TabNet | 23.2 | 0.814 | 0.818 | 0.0 | 0.0 | 0.0 | 0.0 | 0.142 |
| kNN | 24.3 | 0.787 | 0.673 | 0.0 | 0.0 | 0.0 | 7.41 | 0.166 |
| SwitchTab | 27.3 | 0.786 | 0.670 | 0.0 | 0.0 | 0.0 | 0.0 | 0.163 |
| NaiveBayes | 30.2 | 0.678 | 0.154 | 0.0 | 0.0 | 0.0 | 0.0 | 0.283 |
| NCM | 31.3 | 0.679 | 0.174 | 0.0 | 0.0 | 0.0 | 0.0 | 0.281 |

**Table E.7:** Full TALENT Binary classification ($> 10,000$ samples) results. Rank is the average rank among the ordered methods over all datasets. (Normalized) Score is the metric we use to compare methods in Figure 3. In this case score is the mean classification accuracy (1 minus the error in Figure 3). Top-X (%) is the percentage of datasets for which that method is in the top X ranks. The final column is the shifted geometric mean error ($SGM_\varepsilon$).

| Method | Rank | Score | Norm. Score | Top-1 (%) | Top-3 (%) | Top-5 (%) | Top-8 (%) | $SGM_\varepsilon$ |
|---|---|---|---|---|---|---|---|---|
| TabPFN-v2 | **5.02** | **0.864** | **0.947** | **26.9** | **59.1** | **72.0** | **80.6** | **0.104** |
| xRFM | 8.75 | 0.856 | 0.893 | 7.53 | 29.0 | 49.5 | 63.4 | 0.111 |
| ModernNCA | 10.3 | 0.856 | 0.886 | 11.8 | 22.6 | 31.2 | 50.5 | 0.111 |
| CatBoost | 10.3 | 0.845 | 0.867 | 4.30 | 23.7 | 34.4 | 54.8 | 0.119 |
| LightGBM | 10.4 | 0.845 | 0.869 | 7.53 | 18.3 | 35.5 | 54.8 | 0.119 |
| TabR | 10.5 | 0.851 | 0.873 | 4.30 | 17.2 | 30.1 | 43.0 | 0.116 |
| XGBoost | 11.4 | 0.844 | 0.851 | 5.38 | 18.3 | 35.5 | 49.5 | 0.120 |
| RealMLP | 11.7 | 0.850 | 0.864 | 1.08 | 7.53 | 18.3 | 35.5 | 0.116 |
| FTT | 13.3 | 0.836 | 0.813 | 1.08 | 6.45 | 17.2 | 32.3 | 0.124 |
| MLP-PLR | 14.2 | 0.838 | 0.817 | 0.0 | 3.23 | 11.8 | 29.0 | 0.124 |
| RandomForest | 14.3 | 0.836 | 0.824 | 3.23 | 10.8 | 19.4 | 33.3 | 0.129 |
| AutoInt | 15.2 | 0.834 | 0.803 | 0.0 | 3.23 | 5.38 | 16.1 | 0.129 |
| Tangos | 15.4 | 0.834 | 0.800 | 0.0 | 3.23 | 7.53 | 16.1 | 0.131 |
| DCN2 | 15.5 | 0.839 | 0.816 | 1.08 | 2.15 | 9.68 | 21.5 | 0.127 |
| Laplace KRR | 15.7 | 0.836 | 0.801 | 1.08 | 16.1 | 20.4 | 26.9 | 0.133 |
| MLP | 16.0 | 0.836 | 0.797 | 1.08 | 3.23 | 8.60 | 17.2 | 0.132 |
| ResNet | 16.1 | 0.839 | 0.811 | 1.08 | 3.23 | 8.60 | 19.4 | 0.129 |
| TabCaps | 16.2 | 0.835 | 0.804 | 1.08 | 3.23 | 5.38 | 14.0 | 0.130 |
| Excelformer | 16.6 | 0.830 | 0.778 | 0.0 | 1.08 | 7.53 | 16.1 | 0.131 |
| SNN | 16.6 | 0.836 | 0.800 | 1.08 | 3.23 | 4.30 | 7.53 | 0.129 |
| Node | 17.0 | 0.824 | 0.757 | 0.0 | 5.38 | 15.1 | 25.8 | 0.137 |
| PTARL | 17.0 | 0.830 | 0.780 | 0.0 | 1.08 | 3.23 | 8.60 | 0.134 |
| Danets | 18.0 | 0.829 | 0.759 | 0.0 | 3.23 | 6.45 | 14.0 | 0.137 |
| TabTransformer | 19.7 | 0.820 | 0.715 | 1.08 | 1.08 | 6.45 | 12.9 | 0.148 |
| LogReg | 20.1 | 0.819 | 0.723 | 3.23 | 9.68 | 12.9 | 17.2 | 0.146 |
| kNN | 20.8 | 0.811 | 0.677 | 5.38 | 10.8 | 14.0 | 18.3 | 0.158 |
| SVM | 21.6 | 0.816 | 0.698 | 2.15 | 3.23 | 6.45 | 10.8 | 0.150 |
| GrowNet | 22.2 | 0.817 | 0.714 | 0.0 | 0.0 | 0.0 | 2.15 | 0.151 |
| SwitchTab | 24.0 | 0.809 | 0.675 | 0.0 | 0.0 | 0.0 | 0.0 | 0.156 |
| TabNet | 25.8 | 0.802 | 0.633 | 0.0 | 0.0 | 0.0 | 1.08 | 0.157 |
| NaiveBayes | 28.6 | 0.695 | 0.281 | 1.08 | 2.15 | 3.23 | 4.30 | 0.253 |
| NCM | 29.9 | 0.723 | 0.230 | 0.0 | 0.0 | 0.0 | 2.15 | 0.257 |

**Table E.8:** Full TALENT Binary classification ($\leq 10,000$ samples) results. Rank is the average rank among the ordered methods over all datasets. (Normalized) Score is the metric we use to compare methods in Figure 3. In this case score is the mean classification accuracy (1 minus the error in Figure 3). Top-X (%) is the percentage of datasets for which that method is in the top X ranks. The final column is the shifted geometric mean error ($SGM_\varepsilon$).

| Dataset | xRFM | XGB | CatBoost | LGBM | RealMLP | MLP-PLR | MLP-RTDL | ResNet-RTDL |
|---|---|---|---|---|---|---|---|---|
| Airlines_DepDelay_10M | 0.9818 | 0.9813 | 0.9796 | 0.9798 | **0.9786** | 0.9795 | 0.9824 | 0.9818 |
| Allstate_Claims_Severity | **0.6489** | 0.6547 | 0.6510 | 0.6530 | 0.6495 | 0.6537 | 0.6557 | 0.6537 |
| black_friday | 0.6881 | 0.6807 | 0.6792 | **0.6787** | 0.6859 | 0.6862 | 0.6929 | 0.6892 |
| Buzzinsocialmedia_Twitter | **0.2080** | 0.2134 | 0.3147 | 0.2789 | 0.2566 | 0.2553 | 0.2840 | 0.2906 |
| nyc-taxi-green-dec-2016 | **0.5834** | 0.6649 | 0.6567 | 0.6489 | 0.6142 | 0.6523 | 0.6657 | 0.6365 |
| wave_energy | **0.0020** | 0.0918 | 0.0499 | 0.0821 | 0.0024 | 0.0073 | 0.0254 | 0.0434 |
| Yolanda | **0.7816** | 0.8012 | 0.8094 | 0.7970 | 0.7869 | 0.7897 | 0.7927 | 0.7856 |

**Table E.9:** Meta-test regression datasets with more than 70,000 samples. Error reported is nRMSE averaged over five train/test splits using `pytabkit`.

| Dataset | xRFM | XGB | CatBoost | LGBM | RealMLP | MLP-PLR | MLP-RTDL | ResNet-RTDL |
|---|---|---|---|---|---|---|---|---|
| airlines | 0.3341 | 0.3292 | 0.3315 | **0.3287** | 0.3344 | 0.3342 | 0.3342 | 0.3339 |
| covertype | **0.0257** | 0.0420 | 0.0612 | 0.0333 | 0.0280 | 0.0364 | 0.0404 | 0.0385 |
| Higgs | 0.2640 | 0.2576 | 0.2576 | 0.2549 | 0.2473 | 0.2528 | 0.2515 | **0.2435** |
| jannis | **0.2701** | 0.2799 | 0.2816 | 0.2779 | 0.2702 | 0.2764 | 0.2865 | 0.2795 |
| MiniBooNE | 0.0539 | 0.0529 | 0.0538 | 0.0525 | **0.0484** | 0.0504 | 0.0503 | 0.0488 |
| numerai28.6 | 0.4778 | 0.4812 | 0.4790 | 0.4782 | 0.4800 | 0.4775 | **0.4771** | 0.4800 |
| porto-seguro | **0.0380** | **0.0380** | 0.0381 | 0.0381 | **0.0380** | **0.0380** | **0.0380** | **0.0380** |
| dionis | 0.0926 | 0.1219 | 0.1041 | 0.1076 | **0.0887** | 0.1257 | 0.1110 | 0.0907 |
| Fashion-MNIST | 0.0889 | 0.0928 | 0.0969 | **0.0895** | 0.0913 | 0.1064 | 0.1041 | 0.1011 |
| kick | 0.0972 | 0.0965 | **0.0956** | 0.0964 | 0.0976 | 0.0978 | 0.0979 | 0.0970 |

**Table E.10:** Meta-test classification datasets with greater than 70,000 samples. Error reported is classification error averaged over five train/test splits using `pytabkit`.

| Model | Elo (↑) | Norm. score (↑) | Avg. rank (↓) | Harm. mean rank (↓) | #wins (↑) | Improva-bility (↓) | Train time per 1K [s] | Predict time per 1K [s] |
|---|---|---|---|---|---|---|---|---|
| AutoGluon 1.3 (4h) | 1779 | 0.673 | 5.2 | 3.2 | 1 | 3.6% | 1734.20 | 7.06 |
| RealMLP (T+E) | 1721 | 0.660 | 6.6 | 5.4 | 0 | 3.1% | 6860.54 | 7.68 |
| ModernNCA (T+E) | 1626 | 0.622 | 9.2 | 2.6 | 3 | 5.4% | 3811.43 | 7.58 |
| LightGBM (T+E) | 1602 | 0.478 | 10.1 | 7.5 | 0 | 6.0% | 686.46 | 5.48 |
| TabDPT (D) | 1575 | 0.515 | 10.9 | 3.8 | 2 | 3.9% | 16.97 | 8.70 |
| xRFM (T+E) | 1563 | 0.492 | 11.5 | 5.3 | 1 | 5.1% | 365.57 | 0.72 |
| CatBoost (T+E) | 1558 | 0.435 | 11.8 | 8.9 | 0 | 5.7% | 2895.38 | 1.32 |
| TabM (T+E) | 1529 | 0.439 | 12.8 | 8.5 | 0 | 4.3% | 4228.53 | 1.19 |
| CatBoost (T) | 1515 | 0.406 | 13.3 | 6.8 | 0 | 5.9% | 2895.38 | 0.07 |
| xRFM (T) | 1504 | 0.406 | 13.8 | 9.7 | 0 | 5.7% | 365.57 | 0.09 |
| LightGBM (T) | 1498 | 0.354 | 13.9 | 10.2 | 0 | 6.8% | 686.46 | 0.74 |
| XGBoost (T+E) | 1470 | 0.328 | 15.2 | 13.9 | 0 | 6.7% | 848.99 | 2.38 |
| XGBoost (T) | 1470 | 0.325 | 15.2 | 13.3 | 0 | 6.7% | 848.99 | 0.47 |
| ModernNCA (D) | 1457 | 0.323 | 15.6 | 11.0 | 0 | 8.1% | 16.07 | 0.29 |
| TabM (T) | 1453 | 0.311 | 16.0 | 12.6 | 0 | 5.2% | 4228.53 | 0.13 |
| RealMLP (T) | 1439 | 0.306 | 16.6 | 13.3 | 0 | 5.9% | 6860.54 | 0.32 |
| CatBoost (D) | 1419 | 0.278 | 17.4 | 12.0 | 0 | 7.9% | 8.35 | 0.09 |
| ModernNCA (T) | 1411 | 0.347 | 17.6 | 6.7 | 0 | 7.9% | 3811.43 | 0.45 |
| TabPFNv2 (T+E) | 1377 | 0.410 | 19.1 | 2.7 | 4 | 4.9% | 3805.62 | 10.41 |
| TabM (D) | 1361 | 0.232 | 20.1 | 16.5 | 0 | 6.9% | 13.90 | 0.12 |
| TabPFNv2 (T) | 1314 | 0.281 | 22.1 | 6.9 | 0 | 6.4% | 3805.62 | 0.26 |
| TorchMLP (T+E) | 1301 | 0.141 | 22.7 | 19.1 | 0 | 8.1% | 4452.11 | 0.85 |
| ExtraTrees (T+E) | 1297 | 0.169 | 23.1 | 16.9 | 0 | 11.1% | 161.73 | 0.78 |
| RealMLP (D) | 1274 | 0.085 | 24.0 | 21.6 | 0 | 8.2% | 23.30 | 1.44 |
| ExtraTrees (T) | 1273 | 0.174 | 24.0 | 16.7 | 0 | 11.4% | 161.73 | 0.12 |
| TabPFNv2 (D) | 1264 | 0.262 | 24.3 | 7.7 | 0 | 7.6% | 2.78 | 0.32 |
| TorchMLP (T) | 1245 | 0.124 | 25.2 | 20.8 | 0 | 8.7% | 4452.11 | 0.09 |
| xRFM (D) | 1236 | 0.149 | 25.8 | 8.6 | 1 | 11.7% | 1.65 | 0.08 |
| LightGBM (D) | 1231 | 0.069 | 25.9 | 24.5 | 0 | 9.7% | 2.03 | 0.30 |
| XGBoost (D) | 1195 | 0.104 | 27.4 | 24.5 | 0 | 10.4% | 2.15 | 0.18 |
| RandomForest (T+E) | 1193 | 0.056 | 27.5 | 25.5 | 0 | 12.1% | 526.17 | 0.77 |
| RandomForest (T) | 1142 | 0.046 | 29.9 | 27.5 | 0 | 12.8% | 526.17 | 0.12 |
| EBM (T+E) | 1133 | 0.122 | 30.0 | 16.4 | 0 | 14.7% | 2124.78 | 0.12 |
| ExtraTrees (D) | 1117 | 0.056 | 30.6 | 27.7 | 0 | 13.0% | 0.42 | 0.06 |
| FastaiMLP (T+E) | 1086 | 0.005 | 31.8 | 30.6 | 0 | 13.1% | 527.21 | 2.83 |
| EBM (T) | 1076 | 0.131 | 32.2 | 9.4 | 1 | 15.3% | 2124.78 | 0.01 |
| TorchMLP (D) | 1075 | 0.015 | 32.4 | 30.3 | 0 | 12.9% | 20.50 | 0.08 |
| FastaiMLP (T) | 1046 | 0.000 | 33.3 | 32.3 | 0 | 13.6% | 527.21 | 0.31 |
| EBM (D) | 1009 | 0.071 | 34.7 | 30.3 | 0 | 16.0% | 7.25 | 0.04 |
| TabICL (D) | 1006 | 0.000 | 34.8 | 34.2 | 0 | 14.2% | 0.63 | 0.06 |
| RandomForest (D) | 1000 | 0.000 | 34.8 | 34.2 | 0 | 14.2% | 0.63 | 0.06 |
| FastaiMLP (D) | 916 | 0.000 | 37.2 | 36.5 | 0 | 17.8% | 3.08 | 0.29 |
| KNN (T+E) | 533 | 0.000 | 43.8 | 43.6 | 0 | 37.3% | 2.25 | 0.15 |
| Linear (T+E) | 485 | 0.000 | 44.3 | 44.2 | 0 | 35.5% | 46.50 | 0.14 |
| KNN (T) | 434 | 0.000 | 44.9 | 44.7 | 0 | 38.0% | 2.25 | 0.03 |
| Linear (T) | 426 | 0.000 | 44.9 | 44.8 | 0 | 35.7% | 46.50 | 0.04 |
| Linear (D) | 312 | 0.000 | 46.0 | 46.0 | 0 | 38.1% | 1.16 | 0.08 |
| KNN (D) | 263 | 0.000 | 46.5 | 46.2 | 0 | 41.6% | 0.04 | 0.02 |

**Table E.11:** Full regression results on TabArena-Lite.

| Model | Elo (↑) | Norm. score (↑) | Avg. rank (↓) | Harm. mean rank (↓) | #wins (↑) | Improva- bility (↓) | Train time per 1K [s] | Predict time per 1K [s] |
|---|---|---|---|---|---|---|---|---|
| AutoGluon 1.3 (4h) | **1521** | **0.586** | **9.0** | **3.0** | **2** | **9.7%** | 4917.81 | 4.05 |
| CatBoost (T) | **1441** | 0.439 | **12.5** | 9.6 | 0 | **11.9%** | 3307.58 | 0.14 |
| CatBoost (T+E) | **1441** | 0.446 | **12.4** | 9.4 | 0 | **11.1%** | 3307.58 | 1.18 |
| TabPFNv2 (T+E) | 1419 | **0.518** | 13.5 | **4.2** | 1 | 13.8% | 2584.13 | 12.37 |
| LightGBM (T+E) | 1418 | 0.393 | 13.2 | 7.5 | 0 | 13.6% | 1280.01 | 4.08 |
| LightGBM (T) | 1405 | 0.381 | 13.9 | 6.9 | 0 | 13.6% | 1280.01 | 1.05 |
| TabM (T+E) | 1387 | 0.405 | 14.8 | 9.7 | 0 | 15.7% | 5568.31 | 1.78 |
| XGBoost (T+E) | 1377 | 0.310 | 15.2 | 13.3 | 0 | 14.0% | 2029.77 | 4.11 |
| TabM (T) | 1373 | 0.401 | 15.5 | 9.1 | 0 | 16.0% | 5568.31 | 0.37 |
| XGBoost (T) | 1373 | 0.317 | 15.6 | 13.1 | 0 | 13.7% | 2029.77 | 1.04 |
| RealMLP (T+E) | 1359 | 0.337 | 16.2 | 5.3 | 1 | 16.8% | 6866.35 | 10.40 |
| TabPFNv2 (D) | 1358 | 0.361 | 16.4 | 4.7 | 1 | 12.5% | 5.48 | 0.35 |
| xRFM (T+E) | 1357 | 0.369 | 16.2 | 7.0 | 0 | 14.8% | 515.01 | 1.68 |
| TabPFNv2 (T) | 1346 | 0.366 | 17.0 | 6.5 | 0 | 14.9% | 2584.13 | 0.41 |
| ModernNCA (T+E) | 1345 | 0.302 | 16.9 | 10.4 | 0 | 15.3% | 6684.65 | 9.59 |
| ModernNCA (T) | 1332 | 0.248 | 17.8 | 12.4 | 0 | 15.6% | 6684.65 | 0.75 |
| RealMLP (T) | 1304 | 0.253 | 19.1 | 15.5 | 0 | 18.4% | 6866.35 | 0.92 |
| CatBoost (D) | 1296 | 0.201 | 19.2 | 16.9 | 0 | 16.1% | 43.10 | 0.25 |
| TabICL (D) | 1293 | 0.325 | 19.9 | **4.5** | 1 | 20.5% | 11.51 | 1.95 |
| TabM (D) | 1293 | 0.284 | 19.9 | 9.4 | 0 | 19.7% | 17.09 | 0.15 |
| ExtraTrees (T+E) | 1286 | 0.289 | 20.2 | 11.0 | 0 | 17.7% | 728.32 | 2.44 |
| xRFM (T) | 1283 | 0.273 | 20.4 | 5.2 | 1 | 16.4% | 515.01 | 0.20 |
| ExtraTrees (T) | 1265 | 0.262 | 21.2 | 11.2 | 0 | 17.3% | 728.32 | 0.36 |
| RandomForest (T+E) | 1260 | 0.321 | 20.9 | 6.1 | 0 | 15.9% | 729.17 | 1.83 |
| TorchMLP (T+E) | 1260 | 0.188 | 21.2 | 17.5 | 0 | 19.6% | 3646.83 | 2.16 |
| TorchMLP (T) | 1218 | 0.156 | 23.9 | 17.8 | 0 | 21.6% | 3646.83 | 0.19 |
| RandomForest (T) | 1190 | 0.241 | 24.9 | 12.1 | 0 | 17.6% | 729.17 | 0.33 |
| XGBoost (D) | 1188 | 0.102 | 24.9 | 21.5 | 0 | 18.1% | 4.93 | 0.59 |
| TabDPT (D) | 1186 | 0.259 | 25.2 | 4.7 | 1 | 23.3% | 33.52 | 20.75 |
| FastaiMLP (T+E) | 1182 | 0.205 | 25.2 | 12.1 | 0 | 24.2% | 2721.87 | 12.59 |
| LightGBM (D) | 1180 | 0.158 | 25.4 | 19.6 | 0 | 21.3% | 5.12 | 0.44 |
| EBM (T+E) | 1168 | 0.166 | 26.0 | 17.1 | 0 | 24.6% | 1471.12 | 0.27 |
| EBM (T) | 1144 | 0.160 | 27.4 | 19.0 | 0 | 24.9% | 1471.12 | 0.03 |
| xRFM (D) | 1129 | 0.094 | 28.4 | 22.1 | 0 | 23.9% | 2.22 | 0.20 |
| ModernNCA (D) | 1110 | 0.051 | 29.1 | 26.8 | 0 | 25.9% | 17.24 | 0.57 |
| FastaiMLP (T) | 1100 | 0.104 | 29.6 | 20.3 | 0 | 25.8% | 2721.87 | 1.08 |
| RealMLP (D) | 1049 | 0.071 | 32.0 | 24.5 | 0 | 26.3% | 26.02 | 4.18 |
| EBM (D) | 1046 | 0.095 | 32.0 | 24.3 | 0 | 27.7% | 6.16 | 0.08 |
| RandomForest (D) | 1000 | 0.000 | 33.9 | 32.0 | 0 | 33.3% | 0.74 | 0.15 |
| TorchMLP (D) | 972 | 0.004 | 35.0 | 33.6 | 0 | 28.5% | 14.37 | 0.36 |
| FastaiMLP (D) | 952 | 0.038 | 35.9 | 31.9 | 0 | 33.4% | 8.37 | 0.66 |
| ExtraTrees (D) | 862 | 0.013 | 38.6 | 35.1 | 0 | 39.1% | 0.76 | 0.15 |
| Linear (T+E) | 752 | 0.000 | 41.8 | 41.5 | 0 | 45.6% | 170.51 | 0.20 |
| Linear (T) | 718 | 0.000 | 42.3 | 41.9 | 0 | 45.8% | 170.51 | 0.13 |
| KNN (T+E) | 714 | 0.000 | 42.4 | 41.0 | 0 | 52.5% | 2.99 | 0.17 |
| Linear (D) | 689 | 0.000 | 42.9 | 42.7 | 0 | 46.9% | 3.89 | 0.16 |
| KNN (D) | 544 | 0.000 | 45.4 | 45.1 | 0 | 69.5% | 0.33 | 0.05 |
| KNN (T) | 541 | 0.000 | 45.6 | 45.5 | 0 | 58.3% | 2.99 | 0.06 |

**Table E.12:** Full multiclass results on TabArena-Lite.

| Model | Elo (↑) | Norm. score (↑) | Avg. rank (↓) | Harm. mean rank (↓) | #wins (↑) | Improva-bility (↓) | Train time per 1K [s] | Predict time per 1K [s] |
|---|---|---|---|---|---|---|---|---|
| AutoGluon 1.3 (4h) | **1501**$_{-32,+32}$ | **0.576** | **9.3** | **3.5** | **4** | **6.7%** | 1102.06 | 1.74 |
| RealMLP (T+E) | **1461**$_{-30,+31}$ | **0.516** | **11.0** | 5.1 | 2 | **8.3%** | 6177.16 | 8.66 |
| TabM (T+E) | **1416**$_{-29,+33}$ | **0.457** | **13.0** | 5.5 | 2 | 9.1% | 2180.12 | 1.13 |
| LightGBM (T+E) | 1398$_{-30,+33}$ | 0.362 | 13.8 | 9.7 | 0 | 11.1% | 328.64 | 0.77 |
| TabICL (D) | 1393$_{-26,+24}$ | 0.427 | 14.0 | **5.0** | 3 | **8.0%** | 8.05 | 2.01 |
| CatBoost (D) | 1380$_{-32,+26}$ | 0.365 | 14.8 | 7.9 | 1 | 10.6% | 3.84 | 0.07 |
| XGBoost (T+E) | 1372$_{-29,+32}$ | 0.358 | 15.1 | 6.7 | 1 | 11.7% | 462.92 | 0.61 |
| CatBoost (T+E) | 1364$_{-32,+33}$ | 0.375 | 15.4 | 8.7 | 0 | 10.8% | 1043.89 | 0.48 |
| ModernNCA (T) | 1361$_{-31,+33}$ | 0.397 | 15.7 | 6.4 | 2 | 10.2% | 3436.74 | 0.41 |
| ModernNCA (T+E) | 1347$_{-30,+29}$ | 0.414 | 16.3 | 5.6 | 2 | 10.4% | 3436.74 | 8.48 |
| CatBoost (T) | 1347$_{-30,+32}$ | 0.331 | 16.6 | 9.9 | 0 | 11.3% | 1043.89 | 0.04 |
| TabM (T) | 1345$_{-40,+23}$ | 0.382 | 16.6 | 7.7 | 0 | 10.3% | 2180.12 | 0.12 |
| XGBoost (T) | 1328$_{-31,+44}$ | 0.305 | 17.2 | 7.6 | 1 | 12.1% | 462.92 | 0.11 |
| LightGBM (T) | 1328$_{-32,+29}$ | 0.288 | 17.4 | 13.1 | 0 | 12.3% | 328.64 | 0.09 |
| TabPFNv2 (T+E) | 1313$_{-34,+35}$ | 0.399 | 18.2 | **3.9** | 5 | 11.8% | 2914.83 | 17.91 |
| xRFM (T+E) | 1310$_{-33,+22}$ | 0.283 | 18.4 | 12.3 | 0 | 12.0% | 247.54 | 0.82 |
| EBM (T+E) | 1296$_{-29,+33}$ | 0.250 | 19.1 | 10.1 | 0 | 14.0% | 852.29 | 0.21 |
| TabM (D) | 1282$_{-27,+35}$ | 0.274 | 19.7 | 11.7 | 0 | 13.4% | 8.00 | 0.12 |
| TorchMLP (T+E) | 1265$_{-25,+36}$ | 0.219 | 20.6 | 13.1 | 0 | 12.3% | 2206.58 | 2.30 |
| TabPFNv2 (T) | 1264$_{-32,+36}$ | 0.296 | 20.9 | 6.5 | 1 | 13.8% | 2914.83 | 0.22 |
| EBM (T) | 1254$_{-32,+30}$ | 0.187 | 21.3 | 12.4 | 0 | 14.7% | 852.29 | 0.02 |
| xRFM (T) | 1251$_{-21,+31}$ | 0.209 | 21.4 | 12.7 | 0 | 13.7% | 247.54 | 0.05 |
| RealMLP (T) | 1248$_{-33,+27}$ | 0.215 | 21.8 | 14.1 | 0 | 13.4% | 6177.16 | 0.26 |
| RealMLP (D) | 1247$_{-33,+34}$ | 0.177 | 21.8 | 14.4 | 0 | 13.3% | 18.75 | 0.99 |
| EBM (D) | 1238$_{-28,+23}$ | 0.191 | 22.3 | 10.3 | 1 | 15.2% | 4.40 | 0.03 |
| TabPFNv2 (D) | 1209$_{-32,+37}$ | 0.241 | 23.9 | 7.0 | 2 | 15.7% | 2.61 | 0.26 |
| FastaiMLP (T+E) | 1206$_{-29,+29}$ | 0.170 | 24.1 | 15.6 | 0 | 15.8% | 561.29 | 4.46 |
| XGBoost (D) | 1198$_{-25,+31}$ | 0.168 | 24.5 | 11.6 | 1 | 15.0% | 1.42 | 0.12 |
| ModernNCA (D) | 1185$_{-27,+28}$ | 0.157 | 25.2 | 11.2 | 1 | 15.8% | 11.15 | 0.31 |
| TorchMLP (T) | 1174$_{-31,+30}$ | 0.156 | 25.7 | 19.3 | 0 | 14.8% | 2206.58 | 0.11 |
| ExtraTrees (T+E) | 1168$_{-27,+25}$ | 0.115 | 26.2 | 19.6 | 0 | 17.3% | 122.90 | 0.58 |
| LightGBM (D) | 1134$_{-29,+27}$ | 0.103 | 28.1 | 23.7 | 0 | 16.1% | 0.93 | 0.09 |
| RandomForest (T+E) | 1132$_{-33,+27}$ | 0.078 | 28.1 | 21.9 | 0 | 18.4% | 171.61 | 0.55 |
| FastaiMLP (T) | 1129$_{-29,+33}$ | 0.124 | 28.2 | 17.7 | 0 | 18.0% | 561.29 | 0.24 |
| TabDPT (D) | 1123$_{-38,+32}$ | 0.152 | 28.7 | 13.2 | 0 | 17.2% | 17.79 | 8.53 |
| ExtraTrees (T) | 1122$_{-30,+27}$ | 0.106 | 28.7 | 16.6 | 0 | 18.9% | 122.90 | 0.07 |
| RandomForest (T) | 1098$_{-24,+29}$ | 0.029 | 29.9 | 27.1 | 0 | 19.1% | 171.61 | 0.05 |
| TorchMLP (D) | 1030$_{-30,+28}$ | 0.052 | 33.2 | 28.2 | 0 | 20.5% | 4.97 | 0.09 |
| RandomForest (D) | 1000$_{-0,+0}$ | 0.032 | 34.5 | 26.6 | 0 | 23.4% | 0.29 | 0.03 |
| FastaiMLP (D) | 998$_{-34,+26}$ | 0.055 | 34.6 | 30.2 | 0 | 21.1% | 2.79 | 0.26 |
| Linear (T+E) | 960$_{-35,+28}$ | 0.037 | 36.3 | 16.6 | 1 | 28.0% | 40.63 | 0.13 |
| xRFM (D) | 943$_{-33,+28}$ | 0.053 | 36.8 | 30.1 | 0 | 24.9% | 1.07 | 0.05 |
| ExtraTrees (D) | 926$_{-41,+28}$ | 0.023 | 37.6 | 33.3 | 0 | 25.4% | 0.18 | 0.03 |
| Linear (T) | 923$_{-36,+28}$ | 0.025 | 37.6 | 27.1 | 0 | 29.0% | 40.63 | 0.05 |
| Linear (D) | 899$_{-31,+32}$ | 0.017 | 38.5 | 35.3 | 0 | 29.8% | 0.95 | 0.08 |
| KNN (T+E) | 722$_{-36,+38}$ | 0.016 | 43.2 | 36.3 | 0 | 47.7% | 2.75 | 0.15 |
| KNN (T) | 691$_{-37,+37}$ | 0.020 | 43.8 | 30.5 | 0 | 48.8% | 2.75 | 0.03 |
| KNN (D) | 421$_{-67,+65}$ | 0.000 | 47.0 | 46.5 | 0 | 56.4% | 0.05 | 0.02 |

**Table E.13:** Full binary classification results on TabArena.

| Method | TabArena | Remaining |
|---|---|---|
| **xRFM** | **0.2380** | **0.3190** |
| TabPFN-v2 | 0.2444 | 0.3323 |
| CatBoost | 0.2818 | 0.3424 |
| RealMLP | 0.2831 | 0.3336 |
| LightGBM | 0.3065 | 0.3580 |
| FT-Transformer | 0.3092 | 0.3963 |
| XGBoost | 0.3168 | 0.3596 |
| AutoInt | 0.3200 | 0.4083 |
| TabR | 0.3241 | 0.3692 |
| Random Forest | 0.3246 | 0.3959 |

**Table E.14:** Performance of the best ten regression methods on the TALENT benchmark restricted to the datasets also included in TabArena (first column). In the second column, performances of the same methods are shown for the remaining datasets not included in TabArena. The value reported is the shifted geometric mean of the error, with the best method of each subset is in bold.

| Method | TabArena | Remaining |
|---|---|---|
| **xRFM** | **0.1485** | 0.1062 |
| TabPFN-v2 | 0.1487 | **0.1021** |
| CatBoost | 0.1502 | 0.1150 |
| LightGBM | 0.1503 | 0.1154 |
| ModernNCA | 0.1510 | 0.1060 |
| TabR | 0.1515 | 0.1093 |
| RealMLP | 0.1522 | 0.1117 |
| MLP-PLR | 0.1524 | 0.1203 |
| XGBoost | 0.1524 | 0.1156 |
| FT-Transformer | 0.1530 | 0.1211 |

**Table E.15:** Performance of the best ten binary classification methods on the TALENT benchmark restricted to the datasets also included in TabArena (first column). In the second column, performances of the same methods are shown for the remaining datasets not included in TabArena. The value reported is the shifted geometric mean of the error, with the best method of each subset is in bold.

| Method | TabArena | Remaining |
|---|---|---|
| TabPFN-v2 | **0.0737** | 0.1107 |
| XGBoost | 0.0763 | 0.1408 |
| LightGBM | 0.0775 | 0.1436 |
| **xRFM** | 0.0791 | 0.1185 |
| CatBoost | 0.0803 | 0.1312 |
| RealMLP | 0.0804 | 0.1160 |
| ModernNCA | 0.0874 | **0.1097** |
| TabR | 0.0875 | 0.1152 |
| FT-Transformer | 0.0939 | 0.1356 |
| ExcelFormer | 0.0955 | 0.1407 |

**Table E.16:** Performance of the best ten multiclass classification methods on the TALENT benchmark restricted to the datasets also included in TabArena (first column). In the second column, performances of the same methods are shown for the remaining datasets not included in TabArena. The value reported is the shifted geometric mean of the error, with the best method of each subset is in bold.

| Method | Regression (n=70) | Binary (n=78) | Multiclass (n=52) |
|---|---|---|---|
| xRFM | 0.3393 | 0.1138 | 0.1203 |
| XGBoost | 0.3763 | 0.1246 | 0.1383 |
| CatBoost | 0.3615 | 0.1240 | 0.1310 |
| Portfolio (xRFM or XGBoost) | 0.3425 | 0.1144 | 0.1177 |
| Portfolio (xRFM or CatBoost) | **0.3379** | **0.1137** | 0.1179 |
| Portfolio (xRFM, XGBoost, or CatBoost) | 0.3418 | 0.1141 | **0.1173** |

**Table E.17:** Portfolios of xRFM with CatBoost / XGBoost on the TALENT benchmark (restricted to datasets where validation results are provided for XGBoost and CatBoost). Portfolio is constructed by taking the best of the listed methods on the validation dataset and evaluating on the test set. Reported values are SGM of the errors with the lowest error bolded.

# F ADDITIONAL FIGURES

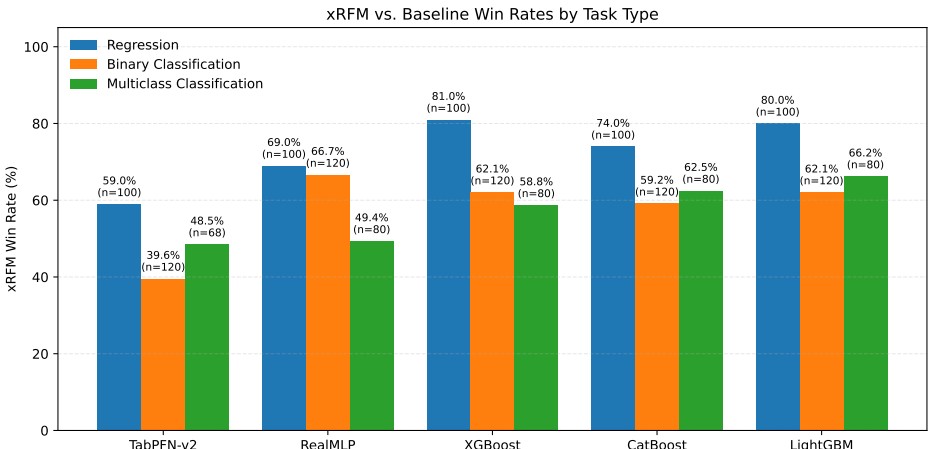

**Figure F.1:** Win-rate of xRFM versus other methods on the TALENT benchmark.

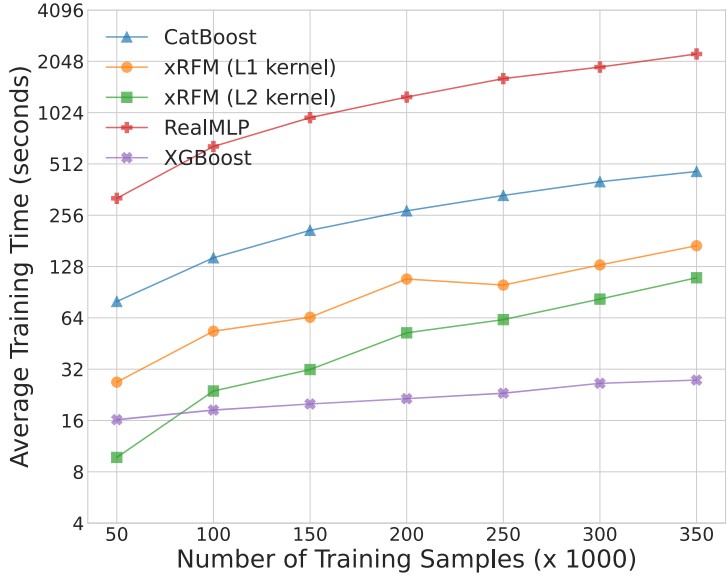

**Figure F.2:** Runtime comparison as a function of the number of training samples on the covertype dataset. Here, L1 kernel refers to the $L_p^p$ kernel.

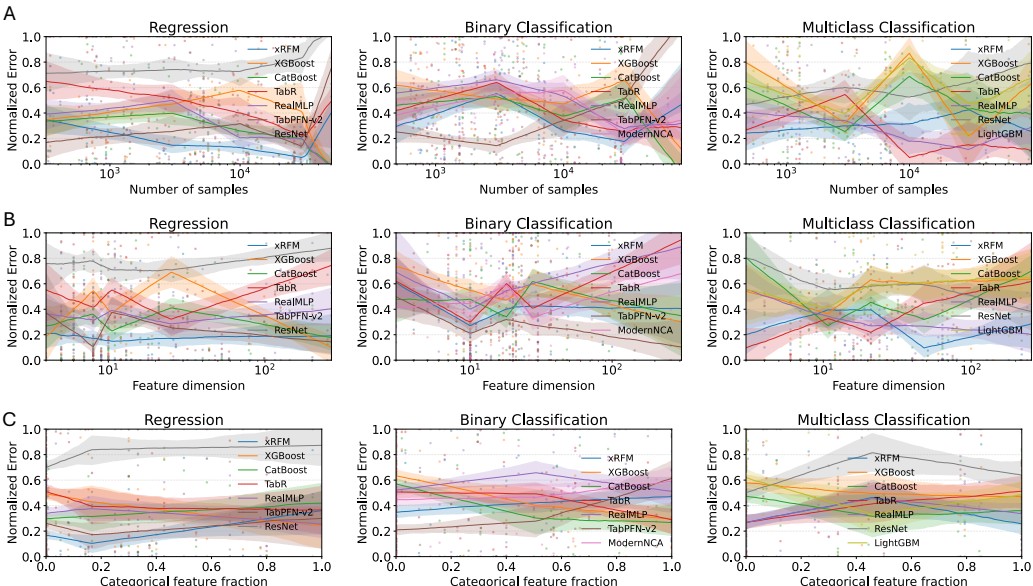

**Figure F.3:** Evaluation of the normalized errors across dataset metafeatures for the the top methods. A piece-wise linear fit is shown for (A) number of samples, (B) feature dimension, and (C) the fraction of features that are categorical versus the normalized error. Individual datasets are shown as dots.

