# OpenReview forum: "xRFM: Accurate, scalable, and interpretable feature learning models for tabular data"
_ICLR.cc/2026/Conference — ICLR 2026 Poster_

### Official Review · Reviewer_LmzM · 2025-10-26

**Soundness:** 3
**Presentation:** 3
**Contribution:** 3
**Rating:** 6
**Confidence:** 3

**Summary:**

This paper introduces a new architecture for tabular prediction tasks called xRFM. Specifically, xRFM combines feature learning kernel machines with a tree structure to both adapt to the local structure of the data. Empirically, the authors showed that xRFM outperforms various baselines in tabular regression tasks and shows competitive results in tabular classification tasks.

**Strengths:**

1. The proposed method is fast, scalable, efficient, and also accurate.

2. The authors have done comparative experiments on a total of 300 datasets. This is a really big contribution to the tabular learning society and ICLR.

3. xRFM is a very efficient solution of tabular prediction tasks, while ensuring competitive performance.

4. The paper is well written.

5. Tabular prediction tasks are challenging and important problems in ML society. Moreover, I highly agree that this field is still dominated by simple GBDTs and some new architectures should be proposed progressively.

**Weaknesses:**

1. Can the authors provide the win rate of xRFM compared to competitive baselines like TabPFN-v2, RealMLP, etc?

2. Is xRFM also effective for few-shot tabular predictions tasks, where the number of class-per samples is just 1 or 5?

3. Are you going to open-source this project?

**Questions:**

See the above Weaknesses.

---

> ### Author Response · Authors · 2025-11-21
> **Author Response to to Reviewer LmzM**
>
> We thank the reviewer for their feedback and for recognizing the speed, scalability, and accuracy of our method. Based on reviewer suggestions, we have provided a number of new experimental results (see our global comment for a summary). We respond to the reviewer’s individual questions and suggestions below.
>
> >Can the authors provide the win rate of xRFM compared to competitive baselines like TabPFN-v2, RealMLP, etc?
>
> We provide a new plot showing xRFM exhibits a favorable win-rate against other strong tabular baselines, mirroring the results based on SGM (Fig. F.1).
>
> >Is xRFM also effective for few-shot tabular predictions tasks, where the number of class-per samples is just 1 or 5?
>
> To answer the reviewer’s question, we generate new plots showing the effect of sample size on the relative performance of xRFM (new Fig. F.3). We find that xRFM maintains strong performance even when the number of samples is small, a setting which may be similar to few-shot learning.
>
> >Are you going to open-source this project?
>
> We have already open-sourced the project and xRFM has been integrated into three other major libraries, which we do not share here for anonymity.  We have also provided anonymized code for the project in our original submission in the supplemental material.

---

> ### Comment · Reviewer_LmzM · 2025-11-24
>
> Thanks for the rebuttal and the additional effort for new experiments. Consequently, I raised the score, and believe this paper can be a strong contribution to ICLR & tabular learning community.

---

### Official Review · Reviewer_wAt5 · 2025-10-28

**Soundness:** 4
**Presentation:** 4
**Contribution:** 3
**Rating:** 8
**Confidence:** 4

**Summary:**

The paper proposes xRFM, a model that uses the Average Gradient Outer Product (AGOP) to identify directions of maximal predictive variation, recursively splitting the data and fitting local kernel ridge regressors in each leaf. Overall, this is a solid paper that effectively demonstrates the usefulness of AGOP for tabular data.

**Strengths:**

This is a solid paper that effectively demonstrates the usefulness of AGOP for tabular data.

**Weaknesses:**

See Questions below.

**Questions:**

I have a few minor points for the authors to address:

1. In Equation (2), what happens if the gradients point in opposite directions or are orthogonal? Could this cause AGOP to fail or weaken its signal?

2. Please investigate or discuss why the proposed method performs well on TALENT, but not better than fine-tuned CatBoost on TabArena.

3. Could gradient boosting be combined with xRFM to further improve performance? A small-scale study or discussion on this would be helpful.

4. Figure 2 nicely illustrates how xRFM captures feature interactions. For this method, is it possible to derive a feature importance ranking similar to that in XGBoost? Please consider adding a short discussion.

5. Please consider performing a post-hoc meta-analysis to identify which types of datasets favor xRFM and which ones favor existing methods such as PFN or CatBoost.

---

> ### Author Response · Authors · 2025-11-21
> **Author Response to Reviewer wAt5**
>
> We thank the reviewer for their feedback and for recognizing the utility of our approach for predictive modeling on tabular datasets. Based on reviewer suggestions, we have provided a number of new experimental results (see our global comment for a summary). We respond to the reviewer’s individual questions and suggestions below.
>
> >In Equation (2), what happens if the gradients point in opposite directions or are orthogonal? Could this cause AGOP to fail or weaken its signal?
>
> If the gradients point in opposite directions, their outer product is the same, because $(-\nabla f)(-\nabla f)^\top = \nabla f (\nabla f)^\top$. If the gradients are orthogonal, they will both be represented in the AGOP matrix $M$. You can visualize this by looking at $v^T M v = \sum (v^T \nabla f)^2$. If v is parallel to one $\nabla f$ but orthogonal to another one, the orthogonal one will not affect the result of $v^\top M v$.
>
> >Please investigate or discuss why the proposed method performs well on TALENT, but not better than fine-tuned CatBoost on TabArena.
>
> Per the reviewer’s suggestion, we have conducted a number of new experiments to investigate the difference in performance on these two benchmarks.  Our new results show that xRFM outperforms fine-tuned CatBoost on TabArena when evaluating methods based on normalized error as opposed to the ELO metric in TabArena (updated Fig. 3G).  We further demonstrate that the difference in results between TabArena and TALENT cannot be explained by dataset selection alone. In particular, we show xRFM is the best overall method on regression and binary classification on the subset of datasets in TALENT that is also in the TabArena benchmark (new Tables E.14 and E.15). Other factors that could contribute to the explanation are the use of random search (instead of Optuna), cross-validation, and stronger (more expensive) baselines in TabArena. Moreover, for classification, TabArena uses different metrics (AUC and log-loss), where the log-loss could potentially be improved using post-hoc calibration. We include this discussion in new Appendix C.
>
> >Could gradient boosting be combined with xRFM to further improve performance? A small-scale study or discussion on this would be helpful.
>
> We thank the reviewer for the suggestion.  We provide new experiments that show including xRFM in a portfolio with XGBoost and CatBoost gives improved performance over any of the three single models alone on the TALENT benchmark (new Table E.17).
>
> >Figure 2 nicely illustrates how xRFM captures feature interactions. For this method, is it possible to derive a feature importance ranking similar to that in XGBoost? Please consider adding a short discussion.
>
> Yes, we can derive feature importances from the diagonal of the AGOP matrices at the leaf RFM models. These give the $E_z[ (\frac{df}{dx(i)}(z))^2]$ over the training data.  These scores (via connection through the Neural Feature Ansatz from [1]) have been linked to independence test statistics between input coordinates and labels [Theorem 2, 2].
>
> [1] Radhakrishnan, A., Beaglehole, D., Pandit, P., & Belkin, M. (2024). Mechanism for feature learning in neural networks and backpropagation-free machine learning models. Science, 383(6690), 1461-1467.
>
> [2] Radhakrishnan, A., Jain, Y., Uhler, C., & Lander, E. S. (2025). Efficiently quantifying dependence in massive scientific datasets using InterDependence Scores. Proceedings of the National Academy of Sciences, 122(34), e2509860122.
>
> >Please consider performing a post-hoc meta-analysis to identify which types of datasets favor xRFM and which ones favor existing methods such as PFN or CatBoost.
>
> We provide new meta-analyses showing the effects of feature dimension, number of samples, and fraction of categorical variables on the relative performance of different predictive models (new Fig. F.3).  These plots do not demonstrate a clear relationship between the performance of xRFM and dimensional properties of the data, suggesting that xRFM is an overall well-balanced tabular predictive model.

---

### Official Review · Reviewer_7Jyu · 2025-10-28

**Soundness:** 3
**Presentation:** 3
**Contribution:** 2
**Rating:** 6
**Confidence:** 4

**Summary:**

This paper introduces **xRFM**, a novel algorithm for learning from tabular data that integrates **feature-learning kernel machines** with **adaptive tree-based data partitioning**. The key idea is to build a binary tree that recursively splits the data along informative directions derived from the **Average Gradient Outer Product (AGOP)**, training a *Recursive Feature Machine (RFM)* at each leaf. xRFM achieves **log-linear training complexity** and **logarithmic inference time**, scaling effectively to very large datasets (up to 500K samples). It is evaluated extensively on **TALENT**, **TabArena-Lite**, and **Meta-Test** benchmarks, outperforming 31 baselines (including GBDTs and TabPFN-v2) on regression tasks and remaining competitive on classification tasks.

**Strengths:**

* The combination of **kernel-based feature learning** (via RFM) with **tree-based local partitioning** and echanism based on **AGOP decomposition** is conceptually novel and elegant.

* The paper demonstrates rigorous experimental validation across **three major benchmarks** and **hundreds of datasets**, establishing strong empirical credibility. The comparisons with baselinse are comprehensive, with fair baselines including recent foundation models like **TabPFN-v2** and **RealMLP**.
* The **scaling analysis** (O(n log n) training, O(log n) inference) is clearly supported by empirical runtime curves (Fig. 5).
* The method maintains theoretical ties to kernel learning and AGOP-based supervised PCA, providing a sound foundation for interpretability and feature relevance analysis.
* The paper is very well written, with clear motivation, detailed algorithmic descriptions (Algorithms A.1–A.5), and informative figures (especially Fig. 1,2).
* The paper addresses a long-standing gap: scalable, interpretable, and high-performing models for **tabular data**. The reported performance and efficiency make xRFM a viable candidate for large-scale deployment in applied ML contexts (e.g., finance, healthcare, industrial analytics).

**Weaknesses:**

* While the empirical results are compelling, the paper lacks a **formal analysis** of convergence or generalization bounds for the tree-partitioned RFM structure.

* The paper could better isolate the contributions of individual components:
    * How much performance gain is from **tree-based scaling**? For example a comparison with normal RFM can show it.
    * How much performance gain is from **AGOP**? If using normal tree splitting, how much will the performance drop? What if using other splitting method?

**Questions:**

See weakness

---

> ### Author Response · Authors · 2025-11-21
> **Author Response Reviewer 7Jyu**
>
> We thank the reviewer for their feedback and for recognizing the strengths of our method and results. Based on reviewer suggestions, we have provided a number of new experimental results (see our global comment for a summary). We respond to the individual questions and feedback below.
>
> >While the empirical results are compelling, the paper lacks a formal analysis of convergence or generalization bounds for the tree-partitioned RFM structure.
>
> While we do not provide a theoretical analysis for the tree splitting itself, we note that the leaf RFM models have theoretical guarantees in the multi-index model and low-rank matrix recovery settings [1, 2].  Even these theoretical results are far beyond what is available for competing state-of-the-art models on tabular data (e.g., complex neural network architectures, such as TabPFN - published in Nature, TabM, RealMLP, ModernNCA, TabICL, etc.), which have, as of yet, no formal analysis of convergence or generalization.
>
> [1] Zhu, Davis, Drusvyatskiy, Fazel. “Iteratively reweighted kernel machines efficiently learn sparse functions”, arXiv pre-print, 2025.
>
> [2] Radhakrishnan, Belkin, Drusvyatskiy. “Linear Recursive Feature Machines provably recover low-rank matrices” PNAS, 2025.
>
> >How much performance gain is from tree-based scaling? For example a comparison with normal RFM can show it.
>
> We show in Fig. 4 that xRFM significantly outperforms the original RFM on the TALENT benchmark. We also present new results that tree scaling enables xRFM to maintain its performance lead over the original RFM on the largest datasets from TALENT (new Table E.3). Our new results show that the normalized errors of xRFM are lower than RFM on this large subset. xRFM enjoys this improvement over the original RFM, while also having log-linear computational complexity in the number of samples. Here, the original RFM uses a previous state-of-the-art scalable training technique (EigenPro), which is based on pre-conditioned gradient descent in function space, and has quadratic computational complexity. In our experience, the Eigenpro solver used with RFM is much more difficult to scale to large dataset sizes and can fail to converge for kernels with $p \neq 2$.
>
> >How much performance gain is from AGOP? If using normal tree splitting, how much will the performance drop? What if using other splitting method?
>
> The AGOP significantly improves the leaf RFM over just the kernel ridge regression model (Fig. 4A). Further, supervised splits with the AGOP are significantly better than splitting using unsupervised methods such as PCA, even with ensembling leaf RFM models (see our new Tables E.1 and E.2). Our Fig. 2 provides a simple synthetic dataset demonstrating the advantage of AGOP-based splits. The top eigenvector of the AGOP detects the $x_0$ feature, which partitions the data optimally for the task.

---

> > ### Comment · Reviewer_7Jyu · 2025-11-24
> >
> > Thank you for your responses and my concerns are addressed. I will keep my score.

---

### Official Review · Reviewer_XCAi · 2025-10-31

**Soundness:** 3
**Presentation:** 4
**Contribution:** 3
**Rating:** 6
**Confidence:** 4

**Summary:**

This paper addresses the limitations of traditional tabular data modeling in terms of scalability and feature learning capabilities and proposes a new model, xRFM. This model aims to simultaneously possess (1) local feature learning capabilities, (2) interpretability, and (3) computational efficiency that can be scaled to very large datasets, thereby surpassing existing gradient boosted decision trees (GBDTs) and the recent tabular data base model in both regression and classification tasks.

**Strengths:**

S1. Combining RFM with tree-based data partitioning and using AGOP main directions for supervised splitting, this combination of local feature learning and scalable kernel methods is novel in tabular ML.

S2. The method's detailed derivation is clear, and the theoretical background and splitting criteria for AGOP are clearly referenced and supported. Technical improvements, such as kernel parameter space exploration, categorical variable optimization, and bandwidth adaptation, are detailed.

S3. Experiments covering ultra-large datasets (>500,000 samples) demonstrate computational scalability. Multiple benchmarks validate its advantages in regression tasks.

S4. Feature interpretation is natively supported, eliminating the need for external tools. Locally relevant features can be analyzed directly from the AGOP of each leaf node.

**Weaknesses:**

W1. Based on experimental tables (such as the TALENT binary and multi-classification results), xRFM is only competitive in most classification tasks, rather than significantly leading. Significant improvements in a single regression domain do not fully demonstrate the model's versatility.

W2. The results lack fine-grained analysis comparing different model families. Although benchmark rankings are reported, there is a lack of grouped performance analysis for model families (Tree-based, Kernel-based, and NN-based), which fails to clearly demonstrate in which structural scenarios xRFM excels. The charts focus on overall rankings and do not show the correlation between different task attributes and performance.

**Questions:**

Q1. Is the AGOP splitting criterion stable in the case of high-dimensional sparse features? Has a comparison been conducted with splitting methods based on axis information gain?

Q2. What is the main bottleneck that causes xRFM's performance to lag behind in multi-classification tasks? Is it the tree splitting strategy, the Leaf RFM structure, or insufficient hyperparameter tuning?

Q3. If xRFM is combined with TabPFN-v2 (as a backend or feature extractor), can the gap in classification tasks be narrowed?

---

> ### Author Response · Authors · 2025-11-21
> **Author response to Reviewer XCAi**
>
> We thank the reviewer for their feedback and for recognizing the strengths in the new paradigm, presentation, large-scale experiments, and interpretability. Based on reviewer suggestions, we have provided a number of new experimental results (see our global comment for a summary). We respond to the individual questions and feedback below.
>
> >W1. Based on experimental tables (such as the TALENT binary and multi-classification results), xRFM is only competitive in most classification tasks, rather than significantly leading. Significant improvements in a single regression domain do not fully demonstrate the model's versatility.
>
> We highlight that in addition to being the single best model for regression on the 100 regression datasets from TALENT, xRFM is the best method on around 7-30% of classification datasets on TALENT (see Tables E.5-E.8, the Top-1% column).
>
> Moreover, automated ML methods like AutoGluon and benchmarks like TabArena have clearly shown that the strongest results come from a diverse ensemble of methods, and we expect xRFM to contribute to such an ensemble as it is conceptually quite different from most other tabular methods. As a demonstration of this, we show that one can improve over xRFM or GBDTs alone by combining xRFM, CatBoost and XGBoost in a model portfolio (new Table E.17).
>
> xRFM also beats many competitors in terms of training and inference time (new Fig. 3 in our updated submission). Another important criterion for tabular data methods in practice is interpretability, which is offered natively by xRFM through the AGOP matrices of its leaf RFM models (Fig. 7).
>
> In contrast to most other tabular methods that build on a large amount of research for deep learning or tree-based methods, xRFM promotes a kernel-based approach, which is unique in the applied tabular data community and may inspire future methods.
>
>
> >W2. The results lack fine-grained analysis comparing different model families. Although benchmark rankings are reported, there is a lack of grouped performance analysis for model families (Tree-based, Kernel-based, and NN-based), which fails to clearly demonstrate in which structural scenarios xRFM excels. The charts focus on overall rankings and do not show the correlation between different task attributes and performance.
>
> To answer the reviewer’s question about whether there is any correlation between different task attributes and performance, we have added new plots that show the relation between average rank and number of samples, number of features, and the fraction of categorical features (new Fig. F.3).  From what we can see, no strong trends can be concluded, suggesting that xRFM is an overall well-rounded method. In fact, as foundation models suffer on large datasets, xRFM becomes the best overall method in that regime.
>
> >Q1. Is the AGOP splitting criterion stable in the case of high-dimensional sparse features? Has a comparison been conducted with splitting methods based on axis information gain?
>
> Regarding the first question regarding stability of splits: As is the case for most tree construction methods, we expect that the specific split found by AGOP could be quite sensitive to small variations (e.g., the top eigenvector of a matrix is unstable whenever the largest two eigenvalues are close). However, this does not necessarily mean that the final score is very sensitive to the choice of split direction at any given node.
>
> Regarding the second question: In our updated paper, we include a comparison to a tree-based splitting mechanism that splits along the axis where the variance in the leaves is most reduced. This mechanism performs comparably to AGOP-based splitting (new Tables E.1 and E.2).

---

> ### Author Response · Authors · 2025-11-21
> **Additional author response**
>
> >Q2. What is the main bottleneck that causes xRFM's performance to lag behind in multi-classification tasks? Is it the tree splitting strategy, the Leaf RFM structure, or insufficient hyperparameter tuning?
>
> The tree splitting strategy, leaf RFM structure, and hyperparameter tuning are the same across multiclass, binary, and regression. Moreover, TALENT and TabArena only contain a few large datasets on which tree splitting is used. Therefore, these cannot explain the gap in multiclass performance. Instead, we see other points: (1) Kernel methods are naturally suited for learning relatively smooth functions, and perhaps this is generally a better model for regression than for multi-class classification; (2) Neural networks can learn representations that are jointly optimized for all classes while xRFM learns the AGOP jointly for all classes, but the kernel-based predictors are learned independently for each class; (3) On TabArena, the logloss is used for multiclass classification, whereas xRFM optimizes the MSE.
>
> We note there are additional strategies for improving performance of xRFM on multi-class tasks: In a communication with the TabArena team, we have heard of preliminary results that xRFM can benefit substantially from post-hoc calibration methods for the multiclass datasets.
>
> >Q3. If xRFM is combined with TabPFN-v2 (as a backend or feature extractor), can the gap in classification tasks be narrowed?
>
> Yes, we demonstrate that training xRFM on embeddings from TabPFN-v2 closes the gap on classification (see new Fig. F.4). As noted in our response to W1, xRFM can also contribute to the SOTA by being used in an ensemble of multiple methods, which is the approach typically used in automated machine learning methods.

---

### Author Response · Authors · 2025-11-21
**Global comment**

We thank the reviewers for their detailed feedback and evaluation of our submission. We have uploaded a revised submission including a number of additional experimental results addressing reviewer questions and suggestions. Changes from the original text are marked in blue, including, in particular, new Appendices B-E with extended discussion of the new results. We provide a summary of these additional results here, in addition to responses to individual reviews. In particular:

1. We address the difference in ranking between xRFM and CatBoost on TALENT and TabArena benchmarks. We show xRFM outperforms all methods, except for the TabDPT foundation model, on TabArena-Lite on regression, and is among the top three methods for binary classification when evaluating the normalized error instead of the ELO metric (Updated Fig. 3 in the main text). We further show that xRFM maintains its lead over GBDTs on the TabArena subset of datasets from the TALENT benchmark (new Tables E.14 and E.15).

2. We additionally demonstrate that xRFM exhibits a favorable winrate versus RealMLP, GBDTs, and TabPFN-v2 on TALENT (new Fig. F.1).

3. We show that xRFM significantly improves over the original RFM trained using alternative scaling methods on the large datasets from TALENT, while reducing the computational complexity from quadratic to log-linear (new Table E.3).

4. We additionally introduce ensembling over leaf models at prediction time with automatic temperature tuning, and show this technique improves performance on large datasets (new Tables E.1 and E.2).

5. We evaluate various other methods for extracting split directions, demonstrating that supervised split directions have better performance than unsupervised ones (new Tables E.1 and E.2).

6. We produced a number of new plots demonstrating the effect of feature dimension, number of samples, and fraction of categorical variables on the relative performance of different predictive models (new Fig. F.3).

7. We demonstrate that xRFM can be ensembled with GBDTs (CatBoost and XGBoost) to improve performance over either model choice individually on the TALENT benchmark (new Table E.17).

8. We demonstrate that xRFM can be combined with TabPFN-v2 to further improve performance on classification tasks from the TALENT and TabArena benchmarks (new Fig. F.4). We do so by training xRFM on embeddings extracted from TabPFN-v2.

---

### Meta-Review · Area_Chair_grZX · 2025-12-03

**Summary:**

This paper proposes a new type of predictor for tabular classification and regression problems.  In experiments, the authors show that their approach surpasses tabpfn-v2 and GBDT models on regression and matches SOTA models on classification.  Their approach also comes with a natural interpretability method.  The reviewers raised several concerns: (1) lack of fine-grained analysis determining which scenarios the proposed method excels in, (2) lack of formal guarantees, (3) win rates.  The remaining criticisms mostly were minor or represented misunderstandings by the reviewers which have been cleared up.

**Reviewer Concerns:**

(1) and (2) were not really addressed, but I absolutely do not think these are disqualifying at all.  After all, very few papers that produce highly influential methods come with formal guarantees, and formal guarantees are not generally practically informative anyways.

**Reviewer Scores:**

The scores were originally 6, 6, 8, 6.  At least one reviewer indicated that they would further raise their score. Clearly the reviewers vote to accept with an average score of at least 7.

---

### Decision · Program_Chairs · 2026-01-26

Accept (Poster)